behaviour, ecology, evolution

stress-axis programming, social competence, behavioural flexibility, colour discrimination, reversal learning, cichlid

**Author for correspondence:**
Maria Reyes-Contreras
e-mail: maria.reyes-contreras@iee.unibe.ch

# Stress axis programming generates long-term effects on cognitive abilities in a cooperative breeder

Maria Reyes-Contreras and Barbara Taborsky

Division of Behavioural Ecology, Institute of Ecology and Evolution, University of Bern, Wohlenstrasse 50A, CH-3032 Hinterkappelen, Switzerland

MR-C, 0000-0002-0862-7436; BT, 0000-0003-1690-8155

The ability to flexibly adjust behaviour to social and non-social challenges is important for successfully navigating variable environments. Social competence, i.e. adaptive behavioural flexibility in the social domain, allows individuals to optimize their expression of social behaviour. Behavioural flexibility outside the social domain aids in coping with ecological challenges. However, it is unknown if social and non-social behavioural flexibility share common underlying cognitive mechanisms. Support for such shared mechanism would be provided if the same neural mechanisms in the brain affected social and non-social behavioural flexibility similarly. We used individuals of the cooperatively breeding fish *Neolamprologus pulcher* that had undergone early-life programming of the hypothalamic–pituitary–interrenal axis by exposure to (i) cortisol, (ii) the glucocorticoid receptor antagonist mifepristone, or (iii) control treatments, and where effects of stress-axis programming on social flexibility occurred. One year after the treatments, adults learned a colour discrimination task and subsequently, a reversal-learning task testing for behavioural flexibility. Early-life mifepristone treatment marginally enhanced learning performance, whereas cortisol treatment significantly reduced behavioural flexibility. Thus, early-life cortisol treatment reduced both social and non-social behavioural flexibility, suggesting a shared cognitive basis of behavioural flexibility. Further our findings imply that early-life stress programming affects the ability of organisms to flexibly cope with environmental stressors.

## 1. Introduction

Individuals that can flexibly adjust their social and non-social behaviour to different contexts may obtain fitness advantages over individuals expressing fixed behaviours. For instance, in the social domain, individuals flexibly adjusting their social behaviour according to social information, such as own and others' rank and/or a partner's fighting experience [1–3], may solve contests faster by showing less energy-costly behaviour [4]. Social flexibility based on the optimal use of social information is also referred to as social competence [2]. Also, outside the social domain, behavioural flexibility can enhance fitness (e.g. [5]). Individuals can benefit from behavioural flexibility to manage decision-making, for instance when feeding or evading predators [6]. Behavioural flexibility is expected to be especially beneficial when adapting to changeable environments, for instance, to urbanization (review in [6]).

Both social and non-social behavioural flexibility are based on learning and memory (e.g. [2,7]). It remains unclear, however, whether social and non-social cognition are based on shared brain mechanisms [8,9], or whether there are special-purpose mechanisms for the social and the non-social domains (see discussions in [10,11]). In a recent conceptual paper, Varela *et al.* [12] proposed three alternative models on how cognition systems might be organized. First, cognitive lower level traits underlying behavioural flexibility, which include the input,

encoding, storage and retrieval of information, are all domain-general [12]. Second, lower level traits may be specialized for social and non-social information processing, suggesting a modular cognitive system [2,12]. Third, cognitive mechanisms may be mixed, with some cognitive lower level traits being specialized for a domain and others being domain-general [13].

Here we ask whether non-social and social behavioural flexibility, that is, the ability to adjust behaviour to new contexts, is affected in a similar or a different way in a vertebrate after the programming of a key physiological system, the hypothalamic–pituitary-adrenal/interrenal (HPA/HPI) axis (also referred to as 'stress axis'). If there were common effects of early-life programming of the stress axis on social and non-social behavioural flexibility, this would support the existence of a general underlying cognition mechanism [12]. The vertebrate stress axis modulates social behaviour [14–17] and social competence [18–20] during ontogeny. It is also an important determinant of cognition and brain development [21,22]. The vertebrate HPA/HPI axis is regulated by glucocorticoids (GCs) and their receptors, the mineralocorticoid (MRs) and glucocorticoid receptors (GRs) [23,24]. Both receptor types are important to acquire, store, consolidate and retrieve information. For instance, MRs are involved in the initial phase of memory encoding; they increase hippocampal excitability and produce emotional hippocampal long-term potentiation (LTP) reinforcement allowing memory formation in rats [23,25,26]. GRs activated, at moderate GC levels, are important for memory consolidation [27], whereas very high GC levels rather inhibit memory in rodents and humans [23,27] and cognitive flexibility in the non-social domain [28]. Conversely, blocking GRs by the GR antagonist mifepristone [29] enhanced memory consolidation in humans [30] and mice [31].

To answer the question whether social and non-social flexibility are modulated similarly by stress-axis programming, we used the cooperatively breeding cichlid fish *Neolamprologus pulcher* as model system. In adults, manipulation of the HPI axis by applying the GR-blocker mifepristone resulted in a short-term enhancement of social competence [19]. Repeated exposure to cortisol during early life decreased the social competence in these fish [20]. In addition, early-life exposure to cortisol or mifepristone resulted in altered stress-axis programming. Both treatments induced a long-term upregulation of the MR gene and downregulation of the corticotropin-releasing factor gene in the telencephalon of adults [20]. Hence in *N. pulcher*, stress-axis programming modulates the development of social flexibility. However, we do not yet know whether early-life HPI axis programming also affects non-social behavioural flexibility in this fish species. Behavioural flexibility in non-social contexts has been most often evidenced by the ability of animals to override previously formed associations, e.g. by reversal learning [6,7,32]. Here, we exposed *N. pulcher* adults that had been treated with cortisol and mifepristone during early life [20] to a two-colour discrimination task at the age of 1.5 years to test their learning ability followed by a reversal-learning task to test their behavioural flexibility. We predicted that early-life exposure to cortisol will impair performance in these tasks [28], whereas mifepristone exposure will increase performance [30,31]. If the underlying mechanism (i.e. stress-axis programming) that modulates behavioural flexibility is shared between the non-social and social domains, then we predict that stress-axis programming changes behavioural flexibility in a non-social task in a similar way than it did for social flexibility [20].

# 2. Methods

## (a) Study species

*Neolamprologus pulcher* is a cooperatively breeding cichlid fish endemic to Lake Tanganyika, East Africa. Its social groups comprise a dominant breeding pair and related and unrelated subordinate individuals, structured in sized-based linear social hierarchies [33]. All group members engage in frequent and diverse social interactions to establish or maintain the social hierarchy or to jointly defend the territory and juveniles against intruders [34,35]. Subordinates can achieve tolerance by dominants by either showing helping behaviour or by showing submission [36–39]. Socially more competent individuals (i.e. more socially flexible individuals) have a higher propensity to respond to breeder aggression by submission [1,3,18] and are more likely to be accepted as a group member by dominants, which is indispensable for survival in the wild [3,34,37]. The development of social competence is influenced by the social environment present during rearing of individuals, such as presence versus absence of adults [1,3,18,19] or group size [40] during early life.

## (b) Early-life treatments

For our experiment, we used fish that received the following pharmacological treatments during their first two months of life during the study by Reyes-Contreras *et al*. [20] (see methods in [20] and the electronic supplementary material): (i) cortisol (the GC hormone of fish (200 ng ml$^{-1}$) [20]), (ii) mifepristone (a GR blocker of fish (400 ng l$^{-1}$) [29]), or (iii) control treatment. Afterwards, until the learning tests, fish were housed in single-sex aggregations of maximally 60 fish in two 200 l compartments of a 400 l tank (for housing conditions, see the electronic supplementary material).

## (c) Experimental design

Forty-eight fish were tested in the two learning tasks, 16 of each early-life treatment. The sex of the individuals was balanced across treatments; hence, from each early-life treatment eight males and eight females were chosen. The age of the fish during the experiment did not differ significantly between the three early life treatments (mifepristone: 533.1 d ± 28.1 mean ± s.e.; cortisol: 581.4 d ± 24.0; control: 603.3 d ± 23.1; generalized linear mixed effect model comparing control (intercept) versus each early-life treatment: cortisol: estimate ± s.e. = −0.0892 ± 0.0680, $z = -1.31$, $p = 0.19$; mifepristone: estimate ± s.e. = −0.0894 ± 0.0695, $z = -1.29$, $p = 0.2$). During the time of the learning tasks, each individual was housed separately in a 25 l tank, which was equipped with 2 cm of gravel sand, one biological filter, and half of a flowerpot in the back of the tank serving as shelter. They could not see fish in adjacent tanks to prevent (i) that they use social cues to solve a learning task and (ii) that territorial aggression between neighbours interferes with learning. At no point of time, fish showed any signs of stress (freezing behaviour, dark skin spots), and they participated deliberately in all trials. Near the front screen of the tank, we placed the experimental apparatus, a grey PVC-plate with four rows of holes and five holes per row (electronic supplementary material, figure S1a,b). Each fish was first habituated to the presence of the plate, which was left permanently inside the tank. Then, we trained the fish to use the experimental apparatus (section 'Training phase'). Subsequently, fish were exposed to a colour discrimination learning task (section 'Acquisition of colour discrimination'), followed by a reversal-learning task (section 'Reversal of colour discrimination'). The experiments were conducted at the Hasli Ethological Station of the Institute of Ecology and Evolution, University of Bern, Switzerland. All experimental procedures were approved by the

Veterinary Office of the Kanton Bern, Switzerland, licence number BE 93/18.

## (d) Training phase

Individuals were trained to (i) dislodge a green plastic disc covering a hole of the grey PVC-plate and (ii) to eat a food reward hidden below the disc inside a hole following methods in [41]. Three consecutive holes from any of the rows of the plate were selected randomly. Inside the first and the third hole, a small piece of krill was placed, while the hole in the middle was left empty. The training was done stepwise, with increasing difficulty to retrieve the food [41]: two green plastic discs of 15 mm diameter [42] were used to progressively cover the holes, in the following sequence: (first) entirely open, (second) one quarter covered, (third) half covered and (fourth) completely covered. During the training phase, the two green discs were always removable by the fish and thus the food reward was always accessible.

Each individual received a maximum of four trials per day. For the first and second level of difficulty, individuals were allowed 1 h to complete the task. For the third and fourth difficulty level, they were given 45 min to complete the task. Trials with the fourth level of difficulty, with completely covered food holes, were video recorded to assess the time required to dislodge both discs and eat the rewards. During the following trial, this time was allowed as maximum time for a given individual to solve the training task. We continued this procedure iteratively, gradually decreasing the time needed to solve a training trial, until all individuals solved the task in 5 min.

Before each trial, an opaque partition was placed in the middle of the tank, temporarily separating the back half of the tank, containing the focal fish and its shelter, from the frontal half with the experimental apparatus. This allowed the experimenter (M.R.C.) to set-up the task without the focal fish seeing the procedure. At the beginning of each trial, a few drops of water containing the smell of krill was added to the tank to provide an incentive for the fish to search for the food item; then, the opaque partition was lifted, and the trial started. The same procedures were carried out in the colour acquisition and reversal-learning tasks (see below).

## (e) Acquisition of colour discrimination

The fish had to learn to discriminate yellow from blue discs (electronic supplementary material, figure S1c,d). These two colours were chosen because *N. pulcher*, and its closely related congener *Neolamprologus brichardi*, attend to the face of conspecifics during social encounters [43], which contain yellow and blue marks [44] (electronic supplementary material, figure S2). To standardize the distance to reach the discs, we always placed the rewards and discs in the row of the hole-plate closest to the shelter of the fish. Again, two holes were filled with a piece of krill and covered with the discs, and an empty hole was left in between. Either the yellow or the blue disc was rewarded. The rewarded colour was balanced across sex and early-life treatments. The fish could easily dislodge the rewarded disc (electronic supplementary material, figure S1c) by either pushing it away or lifting it, as they learned during the training task. However, they could not move the unrewarded disc, as it was blocked by a plastic knob glued to its bottom side that tightly fitted in the holes of the PVC-plate (electronic supplementary material, figure S1d). This allowed us to place a piece of krill under both the rewarded and the unrewarded disc to control for olfactory cues [41].

In each trial, an individual was allowed up to 5 min to make a choice, dislodge the rewarded disc and eat the food reward. The first disc that was touched by a fish with its mouth was considered as chosen. The trial was terminated by placing the opaque divider between the fish and the hole-plate as soon as the fish had dislodged the rewarded disc and had eaten the reward. In the few cases in which a fish did not eat the food reward within 5 min (i.e. the fish had made a wrong choice or no choice, but did not uncover the reward), the rewarded disc was moved to open the hole halfway. Then, the fish was given one additional minute to retrieve the reward (see [41]). If the fish did not succeed by then, the food item was removed from the hole by tweezers and provided directly to the fish. This protocol assured that per trial one piece of krill was eaten and all fish had a similar satiation and motivation level [41].

Each individual received six trials a day. At the beginning of each experimental day, we used a dice to select the position of the rewarded disc (left or right) for the six trials for each fish separately. We used a pseudo-random rule adjusting the ratio of the rewarded : unrewarded side (left or right) to be at maximum 4 : 2 to avoid that fish were unintendedly trained for a side bias. Thus, when the dice determined the same side for the rewarded colour four times in a row, by rule we placed the reward at the other side for the remaining two trials of a day.

## (f) Reversal of colour discrimination

To test for behavioural flexibility, all the fish that successfully reached the learning criterion in the colour acquisition task were exposed to a reversal-learning task. We followed the exact same procedures and learning criterion described for the acquisition task, except that the rewarded colour was reversed for each individual.

## (g) Learning criterion

To assess whether fish from different early-life treatments differ in their ability to learn, both in the colour acquisition and the reversal-learning tasks, we recorded the number of blocks needed to reach the learning criterion. To reach the learning criterion a fish had to make at least five out of six correct choices (i.e. 80% of correct choices) in two consecutive blocks [41,45]. One block consisted of six trials in which the fish had made a correct or wrong choice. If the fish had not made a choice in a given trial, this trial was not counted when evaluating the criterion. Therefore, one block could last for more than 1 day, i.e. until six trials with choice were done. After two consecutive blocks had passed, the learning criterion was assessed. Only three individuals did not reach the learning criterion after 36 trials and were assumed to not have learned the colour acquisition task. These three fish were excluded from the subsequent reversal-learning task.

## (h) Statistical analyses

The statistical analyses were done using the program R v. 3.5.1 [46]. To test the effect of early life treatment on number of blocks needed to reach the learning criterion in the colour acquisition task and the reversal-learning tasks, Cox regression proportional hazard models were fitted using the package 'survival' [47], and the coefficients were estimated by likelihood ratios [48]. In those models, we included the frailty term 'family of origin' as a random effect [49]. Except the factor 'rearing treatment', factors that did not significantly influence the probability to reach the learning criterion were removed from the model by stepwise elimination.

For the colour acquisition task, the initial model (electronic supplementary material, table S1) included rearing treatment (cortisol, mifepristone or control) and colour of the rewarded disc to test for possible colour preferences. In addition, we included sex and age of the fish as covariates. Age has been previously shown to influence learning in this species [45]. We stepwise backward-deleted age and sex of the focal fish from

**Table 1.** Summary table of the Cox regression proportional hazard model testing the effects of early-life treatment, colour of the rewarded disc and age on the number of blocks needed to reach the learning criterion during the acquisition of the colour discrimination task and the reversal task. (The coefficients were estimated using likelihood ratios. Significant results in italics.)

| | coefficient ± s.e. | $\chi^2$ | p |
|---|---|---|---|
| *acquisition of colour discrimination* | | | |
| rearing treatment (cortisol) | 0.482 ± 0.384 | 1.58 | 0.21 |
| rearing treatment (mifepristone) | 0.739 ± 0.396 | 0.39 | 0.062 |
| rewarded colour (yellow) | 1.178 ± 0.334 | 0.33 | *0.00042* |
| frailty (family) | | 0 | 0.94 |
| *reversal of colour discrimination* | | | |
| rearing treatment (cortisol) | −1.197 ± 0.439 | 7.42 | *0.0065* |
| rearing treatment (mifepristone) | −0.496 ± 0.416 | 1.42 | 0.23 |
| rewarded colour (yellow) | 0.689 ± 0.331 | 4.35 | *0.037* |
| age (days) | −0.00787 ± 0.00211 | 13.93 | *0.00019* |
| frailty (family) | | 0 | 0.94 |

**Table 2.** Summary table of the proportion of hazard assumptions of the models in table 1.

| | $\chi^2$ | p |
|---|---|---|
| *acquisition of colour discrimination* | | |
| rearing treatment (cortisol) | 1.57 | 0.21 |
| rearing treatment (mifepristone) | 0.00083 | 0.98 |
| rewarded colour (yellow) | 2.46 | 0.12 |
| global | 4.36 | 0.23 |
| *reversal of colour discrimination* | | |
| rearing treatment (cortisol) | 0.24 | 0.63 |
| rearing treatment (mifepristone) | 1.08 | 0.29 |
| rewarded colour (yellow) | 0.96 | 0.33 |
| age (days) | 0.95 | 0.33 |
| global | 4.04 | 0.40 |

HR = 0.30). In addition, older individuals needed less time to reach the learning criterion in the reversal task than younger ones (table 1; HR = 0.99). Fish of the early-life mifepristone treatment did not differ from control individuals in the reversal task (HR = 0.61). In both tasks, fish reached the learning criterion faster when the rewarded colour was yellow (table 1; HRs = 3.25 and 1.99, see the electronic supplementary material, table S3). The latency to perform the first choice in the acquisition task, which can be regarded as a measure of the motivation to participate in learning tasks, did not differ between treatments (see the electronic supplementary material, table S4).

## 4. Discussion

We had proposed that if there are common effects of early-life stress-axis programming on behavioural flexibility in the social and non-social domains, this would suggest the existence of a shared cross-domain cognitive mechanism [12]. In support of this proposal, we showed that early-life exposure to cortisol impairs behavioural flexibility in a non-social context (green solid arrows in figure 2) in a cichlid fish species. Previous research revealed that the same treatment, i.e. early-life exposure to cortisol, also hampered social flexibility in these fish [20], and that their social flexibility is based on social learning [1]. This implies that exposure to cortisol impaired social learning in *N. pulcher* (green dashed arrow in figure 2). More generally, our results imply that (i) social and non-social flexibility can share a neural substrate in the brain and (ii) that early-life stress programming affects the ability of organisms to flexibly cope with environmental stressors.

### (a) Effects of cortisol

Contrary to our predictions, fish that received a cortisol treatment early in life did not differ from control individuals in their learning abilities during the colour acquisition task. A previous experiment showed that cortisol can have detrimental effects on learning performance. Rats that had received a cortisol implant for a period of 12 weeks performed poorly in a maze test compared to control animals [51]. However, cortisol-treated fish exhibited reduced behavioural flexibility,

the final model, because they did not significantly influence the number of blocks the fish needed to reach the learning criterion in the acquisition of colour discrimination task (table 1). In the model for the reversal task, the rearing treatment, rewarded colour, sex and age were included in the initial model (electronic supplementary material, table S1). Sex of the focal fish did not significantly affect the probability to reach the learning criterion and was dropped from the final model (table 1). For each fixed factor in the initial and final models, the proportion of hazard assumptions was fulfilled (table 2; electronic supplementary material, table S2).

The effect of early life treatment on the number of blocks needed to reach the learning criterion was plotted as the inverse Kaplan–Meier curves using the 'survminer' package [50] (figure 1a,b). Further, we tested the treatment effects on the latencies to make the first choice (see the electronic supplementary material).

## 3. Results

During the acquisition of colour discrimination, individuals that were treated with mifepristone during early life did not differ significantly in their learning performance from the control treatment ($p = 0.06$; table 1); however, mifepristone treated fish have roughly twice the chance to pass the learning criterion in the next block compared to control fish (hazard ratio (HR) = 2.1; see the electronic supplementary material, table S3). Early-life cortisol treatment and control individuals did not differ in the acquisition task (HR = 1.62). During the reversal-learning task, fish that had received a cortisol treatment during early life needed longer to reach the learning criterion, which indicates a lower behavioural flexibility of this treatment group (table 1; figure 1b;

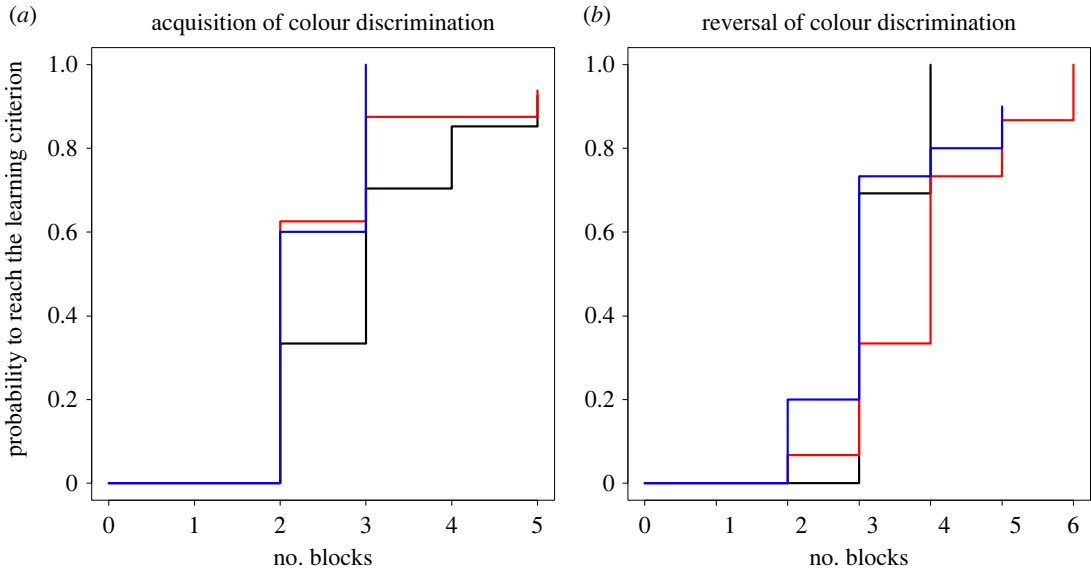

**Figure 1.** Inverse Kaplan–Meier curves showing the results of (*a*) the acquisition of colour discrimination and (*b*) of the reversal-learning task. Black lines: control treatment; red lines: cortisol treatment; blue lines: mifepristone treatment.

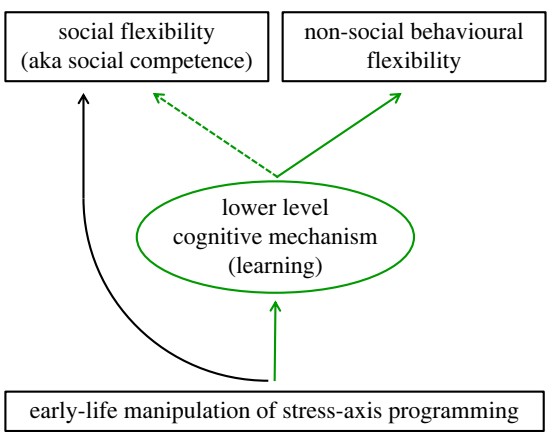

**Figure 2.** Hypothesis resulting from this study that social and non-social flexibility share common lower level cognitive traits. Black: results shown in [20]. Green solid arrows: pathway shown in this study. Dashed green arrow: inference drawn from [1], that social flexibility is based on social learning.

as shown by their poorer performance in a reversal-learning task. This long-term effect was present 1.5 years after the end of the exogenous cortisol treatment. In a previous study, we had reported that early-life cortisol treatment reduced social competence of the fish. It led to an increase of aggressive behaviour during contests over a resource, which extended contest duration while not increasing the chances to win a resource [20]. As aggression is energetically very costly in *N. pulcher* [52], such prolonged contests will increase energy expenditure. Our result that cortisol application impairs behavioural flexibility is in line with findings in other vertebrates. It has been shown that GCs negatively affect neurogenesis, which may be required for fear memory extinction and thus behaviourally flexible adjustment to changing conditions [53,54]. In rats, chronic cortisol treatment significantly impaired cognitive flexibility in the water maze task [28]. Finally, human infants were less able to flexibly adjust their behaviour after exposure to a stressor [54]. After a stressor, infants continued to show a previously rewarded behaviour for longer than non-stressed controls, even if now this behaviour did not produce a reward anymore.

## (b) Effects of mifepristone

Early-life exposure to the GR-blocker mifepristone marginally improved later-life learning abilities compared to control fish, but did not affect behavioural flexibility. The direction of this marginal result agrees with previous studies measuring short-term effects of mifepristone learning abilities in vertebrates [31,55]. For example, a 28-day mifepristone treatment in humans with mood disorders improved their attention and learning ability [30]. Similarly, mifepristone treatment applied 4 h before a memory retrieval test significantly enhanced how humans recalled picture details compared to a placebo group [31].

## (c) Mechanistic link between the stress axis and cognitive abilities

In vertebrates, cognitive performance and the activity of the HPA/HPI stress axis are both modulated by the activation of the two major receptor types of GCs, the GRs and the MRs. These receptors are expressed in the hippocampus and limbic brain areas and also modulate memory formation [56]. The formation of memory requires two processes, LTP, i.e. a persistent strengthening of synapses based on recent patterns of activity [57], and the expression of the cyclic AMP response element-binding protein [56]. Both processes can be modulated by pharmacological manipulations of MRs and GRs [56,57]. Importantly, MR expression was shown to influence the behavioural flexibility in a non-social task in rodents. Rats with an overexpression of MR in the forebrain (independent of their GR expression) had an impaired ability to solve a reversal-learning task [58]. In the fish used in the present study, early-life applications of mifepristone and exogenous cortisol both generated a permanent upregulation of the gene coding for MRs in the telencephalon [20]. Together, these findings suggest that a persistently altered expression of the MRs is involved in the mechanistic link between early-life programming of the stress axis and non-social behavioural flexibly *N. pulcher*. Further research in different vertebrate taxa is needed to show whether MRs are generally involved in the mechanistic basis of vertebrate behavioural flexibility.

## (d) Additional factors influencing cognitive performance

Younger individuals took significantly longer to reach the learning criterion in the reversal task, as reported previously in *N. pulcher* [45]. These results in *N. pulcher* differ from a general tendency found in other vertebrates, in which behavioural flexibility decreased with age (dogs [59], monkeys [60] and rats [61]). Both studies in *N. pulcher* were performed at an age when in nature these fish approach dispersal from their natal groups and/or achieving breeder status, i.e. at around 2–3 years [62], and thus they are about to face a drastic change of their environment. Therefore, it seems plausible that *N. pulcher* exhibit greater flexibility the closer they reach to the age of independent reproduction, whereas in the above-mentioned mammalian experiments, the reported age-related decrease of flexibility reflected senescence.

In both the acquisition and reversal of colour discrimination, *N. pulcher* reached the learning criterion faster when the rewarded colour was yellow, which agrees with previous experiments in this species in non-social colour discrimination tasks [63,64]. Surprisingly, a preference for yellow was absent in a social context, when yellow facial marks were experimentally enhanced [63]. The non-social, ecological relevance of yellow in the natural environment of *N. pulcher* is yet unknown. As we fully balanced the rewarded colour across trials and treatments, the effect of colour on cognitive performance should not systematically bias our results.

## (e) Ultimate implications of a shared cognitive mechanism for behavioural flexibility

In their natural environments, animals encounter numerous non-social and social challenges, during which the ability to express behavioural flexibility may be beneficial. For instance, in the non-social context, individuals should use information about predation risk to adjust anti-predator behaviour [37,65,66]. In the presence of predators, Japanese minnows (*Pseudorasbora parva*) increase foraging activity during nighttime when predation risk is lower [67]. In *N. pulcher*, juveniles adjust their fear behaviour towards heterospecifics based on information about danger learned early in life [66]. In the social domain, animals are exposed to frequent and diverse social interactions with different categories of conspecifics, such as potential mates, group mates, cooperation partners and competitors, which all may have different ranks or resource-holding potentials. In the cooperative breeder *N. pulcher*, maintaining or achieving [68,69] group membership is indispensable for survival [34,70–72]. High social competence enables them to show appropriate social behaviour during the multitude of possible social interactions, which increases their likelihood to be accepted in a group [37].

Evidence from learning experiments for or against the existence of a cognitive mechanism spanning social and non-social domains is thus far equivocal. In laboratory rodents, early social experience modulates social learning but not the performance in non-social learning tasks [73–75], suggesting specialized cognitive mechanisms. In narrow-striped mongooses (*Mungotictis decemlineata*) social learning opportunities in groups affected reversal-learning speed negatively [32]. Conversely, in cooperatively breeding Western Australian magpies (*Cracticus tibicen dorsalis*), performance in several non-social cognition tasks was influenced positively by the complexity of the social environment: performance in these tasks improved with increasing group size [9].

## (f) General implications

Our results have two general implications. First, our results imply that social and non-social flexibility can share a neural substrate in the brain. Evidence of whether social and non-social flexibility are based either on a shared or on specialized, modular cognitive mechanisms has been largely lacking [12] and existing evidence is controversial and rather indirect, resting on studies comparing non-social cognition between individuals exposed to different social conditions [9,45,73]. What could be the advantage of evolving a shared cognitive mechanism? It has been argued that one advantage of a shared cognitive mechanism is that it is less subject to energetic constraints than multiple special-purpose mechanisms, given the finite energy available for parallel brain activity [12]. However, a shared cognitive mechanism used for multiple purposes is likely to be constrained by temporal trade-offs [12], e.g. when attention towards different environmental cues needs to be processed simultaneously by the same cognitive system [12]. Thus, whether shared or special-purpose mechanisms evolve should depend on the relative strength of temporal and energetic constraints present in a species, and thus on a species' ecology.

Second, social and non-social flexibility are impacted by stress programming. Organisms require behavioural flexibility to deal with challenges in their social and non-social environments such as during encounters with competitors, when there are changes in resource availability or when fending off predators. Physiological stress systems, which are present in all organisms including animals, plants and microbes, are integral in coping with such environmental challenges [76]. Programming of the physiological stress system as shown in our study occurs by parental effects or own early experience and has been documented in most major vertebrate taxa, including mammals [77–79], birds [80], amphibians [81] and fishes [20,82–84] as well as invertebrates (reviewed in [85]). Therefore, our finding that non-social and social behavioural flexibility is affected by stress programming points towards an important link between the exposure to environmental stressors, which can lead to stress programming [86,87], and the ability of organisms to flexibly cope with non-social and social stressors.

Ethics. All experimental procedures were approved by the Veterinary Office of the Kanton Bern, Switzerland, licence number BE 93/18.

Data accessibility. Data available from the Dryad Digital Repository: https://doi.org/10.5061/dryad.qfttdz0jr [88].

The data are provided in the electronic supplementary material [89].

Authors' contributions. M.RC.: conceptualization, data curation, formal analysis, visualization, writing—original draft and writing—review and editing; B.T.: conceptualization, data curation, formal analysis, resources, supervision, visualization, writing—original draft and writing—review and editing.

Both authors gave final approval for publication and agreed to be held accountable for the work performed therein.

Conflict of interest declaration. We declare we have no competing interests.

Funding. B.T. was supported by funds from the Swiss National Science Foundation (SNSF, Project 31003A_179208).

Acknowledgements. We are grateful to Evi Zwygart for logistic support. We thank Mias Müller for contributing to data collection, Susana Varela for comments on an earlier draft of this manuscript and Michael Taborsky for discussions.

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
