## [Peer Review File · Proceedings of the Royal Society B: Biological Sciences]

Review History

RSPB-2021-1289.R0 (Original submission)

Review form: Reviewer 1

Recommendation

Major revision is needed (please make suggestions in comments)

Scientific importance: Is the manuscript an original and important contribution to its field?
Excellent

General interest: Is the paper of sufficient general interest?
Good

Quality of the paper: Is the overall quality of the paper suitable?
Good

Is the length of the paper justified?
Yes

Should the paper be seen by a specialist statistical reviewer?
Yes

Do you have any concerns about statistical analyses in this paper? If so, please specify them explicitly in your report.

Yes

It is a condition of publication that authors make their supporting data, code and materials available - either as supplementary material or hosted in an external repository. Please rate, if applicable, the supporting data on the following criteria.

Is it accessible?

Yes

Is it clear?

Yes

Is it adequate?

Yes

Do you have any ethical concerns with this paper?

No

Comments to the Author

This manuscript describes an experiment where cichlid fish were exposed in the first two months of life to six doses of either cortisol (the glucocorticoid/stress hormone in fish), mifepristone (a glucocorticoid blocker), or a control treatment that contained only the carrier of mifepristone. When 18-22 months old, the fish were tested on a colour discrimination and reversal learning task. The authors predicted that the cortisol treatment would impair both colour discrimination and reversal learning performance, while the mifepristone treatment was predicted to enhance performance on both tasks.

The authors found that mifepristone-treated fish performed better on the colour discrimination task, while cortisol-treated fish performed worse on the reversal learning task. Based on these findings and their previous research, the authors conclude that behavioural flexibility in the social (previous study) and non-social domains may share a common underlying cognitive mechanism.

The manuscript is well written and presented and easy to understand. The learning tests follow established protocols, and the results make a significant contribution to the field of animal cognition by suggesting that programming of the HPA axis affects cognitive performance in both social and non-social domains. I enjoyed reading this manuscript and I think the findings are interesting. However, I have some queries regarding the methods, statistical analyses and discussion, which I hope will be helpful to the authors.

Methods:

- If the intensity of facial colour marks is used by these fish to assess conspecific aggressiveness, it's not clear how this relates to distinguishing different colours (rather than colour intensity). Also, if the ecological relevance of the colour discrimination task concerns conspecific interactions, does this task not target performance more closely related to the social domain rather than the non-social domain?
- Please describe at what hours of day the learning tasks were conducted, what the inter-trial interval duration was, whether there were any (overnight) breaks between the colour discrimination and reversal learning tasks, whether the task rewards were the only food the fish were provided with, or whether they were starved beforehand or fed extra afterwards.
- What was the total duration of social isolation for these fish? Could the fish see each other? If not, does this not represent a stressor?
- Why was latency measured for only half the test subjects?
- How was standard length measured? Was the fish taken out of its tank and placed inside

a plastic 'envelope', or was standard length measured from photos/videos? Why was it included in some statistical models – what is this thought to control for?

Statistical Analyses:

- It is unclear why the statistical models of the colour discrimination, reversal learning and latency data all contain different effects; for example, the main text suggests that only the third, latency, model contains 'family of origin' as a random effect, although the tables in the main text and the supplementary materials suggest that 'family' was included in only the first two models. Given that many of the test subjects are from the same broods, it seems that 'family of origin' should be included as a random effect in each statistical model? Similarly, why is 'standard length' included in only the reversal learning model? Also, it seems that sex should be included as a fixed effect in all models?
- Given that each test subject has a score for both colour discrimination and reversal learning performance, it would be interesting to plot individuals' raw scores for each task in the same plot, coloured by early-life treatment. One could visualise and test statistically whether individuals' scores are correlated across tasks (which would support a 'general cognitive ability'), or whether this is only true for the control group, given the contrasting effects of the different treatments.

Discussion:

- Please discuss your finding that longer fish appear to require significantly more trials to reach the learning criterion.
- You predicted that performance on both tasks would be affected by both your treatments. Instead, you found that each treatment affected performance on only one of the tasks, but not the other. Why might this be the case? Please discuss.

Review form: Reviewer 2

Recommendation

Accept with minor revision (please list in comments)

Scientific importance: Is the manuscript an original and important contribution to its field?

Excellent

General interest: Is the paper of sufficient general interest?

Good

Quality of the paper: Is the overall quality of the paper suitable?

Excellent

Is the length of the paper justified?

Yes

Should the paper be seen by a specialist statistical reviewer?

No

Do you have any concerns about statistical analyses in this paper? If so, please specify them explicitly in your report.

No

It is a condition of publication that authors make their supporting data, code and materials available - either as supplementary material or hosted in an external repository. Please rate, if applicable, the supporting data on the following criteria.

Is it accessible?

Yes

Is it clear?

Yes

Is it adequate?

Yes

Do you have any ethical concerns with this paper?

No

Comments to the Author

Review:

Stress axis programming generates long-term effects on cognitive abilities in a cooperative breeder

In this study the authors investigate the performance of a cooperative breeding cichlid from Lake Tanganyika in two learning tests after they experienced an early life programming of the stress axis. During day 10-60 of the fish's development, the offspring were exposed to either i) a cortisol, ii) a mifepristone (a blocker of the glucocorticoid receptors) or iii) a control treatment. Nearly one year later the fish were tested for their learning performance and behavioural flexibility with two learning tests: first a colour discrimination learning test, second a reversal test. In the colour discrimination test fish treated with mifepristone reached the learning criterion quicker than the fish from the control treatment. Further, fish treated with cortisol in their early life required more trials to successfully complete the reversal test compared to the control or mifepristone treated fish.

The authors conclude that early life programming of the stress axis by cortisol or mifepristone influences the learning capability of a highly social cichlid fish. These results together with a previous study, wherein it was shown that the same cortisol treatment led to a reduced social competence in the same species of fish, are evidence that the mechanisms underlying social and non-social flexibility are shaped by the same cognitive mechanisms.

While the search of neuronal mechanisms underlying behavioural flexibility in mammals is well progressed, we still lack a more general understanding of social and non-social flexibility in many other organisms. Therefore, the fascinating results of this well-designed study with a highly social fish as a model species will be of general interest to a wider public.

I have a few, minor comments:

Line 134: The fish used in the experiment were of both sexes. As the treatments were balanced according to sex there is no obligation to investigate a potential sex difference. Nevertheless, I was wondering if the authors ever considered to have a look into potential different abilities between males and females.

Line 142: It is a bit confusing why the authors refer to three tasks here for the first time, as before there were always only two learning tests mentioned. On the first reading it is not clear what is meant by the "motor task". Maybe this sentence can be reformulated to make it clear already here that the "motor task" signifies the training needed to use the experimental apparatus with which the two learning tests are carried out.

Line 157-162: I don't see a problem with the individually flexible time limit based on earlier trials used here. I was just wondering if this procedure reinforces a quick choice of the focal subject on the expense of a higher error rate, which may prolong the training phase.

Line 176: It would be interesting to get some information whether this standardization of the swimming distance by placing the reward always in the row closest to the shelter reduced the error rate and speeded up the choice of the focal fish. Since you used all available rows in the training phase, did you experienced it that the fish was less motivated to search in rows farther

from his shelter?

Line 186: You write "The first disc that was touched by a fish (...) was considered as chosen". Two lines later you write "...and this was done as soon as an individual had dislodged the rewarded disc and had eaten the reward." Does this mean, that you allowed 5 minutes to give the fish the chance to discover the food reward also after a wrong choice?

Line 212: "To reach the learning criterion, a fish had to make at least 10 correct choices in 12 consecutive trials, regardless of non-choice being done." Does this mean the cases where a fish didn't choose, the trial was annulated and not counted for the day? 10 out of 12 means that those 12 choices happened always within two blocks and thus two days, but not more?

Line 237: "Colour" did not significantly influence either model, which is unusual, as several fish species showed clear colour preferences in various colour discrimination tests. However, as far as I know, nobody used blue and yellow for *N. pulcher* before.

Table 1: standard length turned out to be a significant factor in the model of the reversal colour discrimination, however you never discuss this result in the text. Couldn't that be relevant, especially as in fish size is often closely linked with age?

Decision letter (RSPB-2021-1289.R0)

26-Jul-2021

Dear Miss Reyes Contreras:

I am writing to inform you that your manuscript RSPB-2021-1289 entitled "Stress axis programming generates long-term effects on cognitive abilities in a cooperative breeder" has, in its current form, been rejected for publication in *Proceedings B*.

This action has been taken on the advice of referees, who have recommended that substantial revisions are necessary. With this in mind we would be happy to consider a resubmission, provided the comments of the referees are fully addressed. However please note that this is not a provisional acceptance. In particular, you need to make clear the relevance of the study for a broader biological journal such as *Proc B*, rather than a more specialised journal, and the advances of this particular piece of work (rather than the associated earlier work on the same system); acceptance will be dependent on this.

- 1) A 'response to referees' document including details of how you have responded to the comments, and the adjustments you have made.
- 2) A clean copy of the manuscript and one with 'tracked changes' indicating your 'response to referees' comments document.

- 3) Line numbers in your main document.
- 4) Data - please see our policies on data sharing to ensure that you are complying (<https://royalsociety.org/journals/authors/author-guidelines/#data>).

Yours sincerely,
Professor Loeske Kruuk
Editor
mailto: proceedingsb@royalsociety.org

Associate Editor

Comments to Author:

The general question under test in this paper is whether or not the cognitive mechanisms that determine social competence (behavioral flexibility) in animals are shared across social and non-social contexts. The authors propose to test this question by taking an experimental approach to program the vertebrate hypothalamic-pituitary-adrenal/interrenal (HPA/HPI) stress axis during early life in a cooperatively breeding fish (*N. pulcher*), through exposure to cortisol (the glucocorticoid hormone) or mifepristone (a glucocorticoid-receptor blocker), and then by examining the ability of the fish of these treatments (one year after treatment) to learn a colour discrimination task, and then a reversal learning task, relative to fish administered with a control.

The authors report that early-life mifepristone treatment enhanced learning performance, whereas cortisol treatment reduced behavioral flexibility.

The broader question and results are interesting (although the presentation of survival curves was quite drab), and thus the paper was sent to expert review by two referees, who also found the paper interesting, and the experiments well conducted, but raised important issues that bring into question the size of the conceptual advance provided by these experiments. Each referee raised insightful comments that require careful consideration by the authors.

Specifically, the paper ultimately tests and presents one part of a broader study. The abstract and Introduction set up the broader question of whether social and non-social flexibility are regulated similarly by stress axis programming; but the results of social flexibility have been presented previously, and only experiments testing for the effects of pharmacological modifications on non-social flexibility are studied here, with the authors then scaffolding the results to those of the previous studies to make the general conclusions. As a further complication, one of the two referees has questioned whether the experiments conducted here truly target non-social behavioural flexibility, or whether the ecological relevance of the colour discrimination experiment aligns more closely to the social context, than the non-social context, given that the fish use facial colour marks to assess conspecific aggressiveness. This requires careful consideration by the authors.

Other important queries raised by the referees include the question of whether the sex of the fish is likely to have influenced the results; but that this important intrinsic factor is not accounted for in the statistical analyses even though the authors knew the sex of the fish; and there are queries on the statistical models used, and inclusion of random effects. It would appear that 'Family of Origin' would need to be included in the cox proportional hazards models, otherwise the results would be inherently pseudoreplicated.

Reviewer(s)' Comments to Author:

Referee: 1

Comments to the Author(s)

This manuscript describes an experiment where cichlid fish were exposed in the first two months of life to six doses of either cortisol (the glucocorticoid/stress hormone in fish), mifepristone (a glucocorticoid blocker), or a control treatment that contained only the carrier of mifepristone. When 18-22 months old, the fish were tested on a colour discrimination and reversal learning task. The authors predicted that the cortisol treatment would impair both colour discrimination and reversal learning performance, while the mifepristone treatment was predicted to enhance performance on both tasks.

The authors found that mifepristone-treated fish performed better on the colour discrimination task, while cortisol-treated fish performed worse on the reversal learning task. Based on these findings and their previous research, the authors conclude that behavioural flexibility in the social (previous study) and non-social domains may share a common underlying cognitive mechanism.

The manuscript is well written and presented and easy to understand. The learning tests follow established protocols, and the results make a significant contribution to the field of animal cognition by suggesting that programming of the HPA axis affects cognitive performance in both social and non-social domains. I enjoyed reading this manuscript and I think the findings are interesting. However, I have some queries regarding the methods, statistical analyses and discussion, which I hope will be helpful to the authors.

Methods:

- If the intensity of facial colour marks is used by these fish to assess conspecific aggressiveness, it's not clear how this relates to distinguishing different colours (rather than colour intensity). Also, if the ecological relevance of the colour discrimination task concerns conspecific interactions, does this task not target performance more closely related to the social domain rather than the non-social domain?
- Please describe at what hours of day the learning tasks were conducted, what the inter-trial interval duration was, whether there were any (overnight) breaks between the colour discrimination and reversal learning tasks, whether the task rewards were the only food the fish were provided with, or whether they were starved beforehand or fed extra afterwards.
- What was the total duration of social isolation for these fish? Could the fish see each other? If not, does this not represent a stressor?
- Why was latency measured for only half the test subjects?
- How was standard length measured? Was the fish taken out of its tank and placed inside a plastic 'envelope', or was standard length measured from photos/videos? Why was it included in some statistical models - what is this thought to control for?

Statistical Analyses:

- It is unclear why the statistical models of the colour discrimination, reversal learning and latency data all contain different effects; for example, the main text suggests that only the third, latency, model contains 'family of origin' as a random effect, although the tables in the main text and the supplementary materials suggest that 'family' was included in only the first two models. Given that many of the test subjects are from the same broods, it seems that 'family of origin' should be included as a random effect in each statistical model? Similarly, why is 'standard length' included in only the reversal learning model? Also, it seems that sex should be included as a fixed effect in all models?
- Given that each test subject has a score for both colour discrimination and reversal learning performance, it would be interesting to plot individuals' raw scores for each task in the same plot, coloured by early-life treatment. One could visualise and test statistically whether individuals' scores are correlated across tasks (which would support a 'general cognitive ability'), or whether this is only true for the control group, given the contrasting effects of the different treatments.

Discussion:

- Please discuss your finding that longer fish appear to require significantly more trials to reach the learning criterion.
- You predicted that performance on both tasks would be affected by both your treatments. Instead, you found that each treatment affected performance on only one of the tasks, but not the other. Why might this be the case? Please discuss.

Referee: 2

Comments to the Author(s)

Review:

Stress axis programming generates long-term effects on cognitive abilities in a cooperative breeder

In this study the authors investigate the performance of a cooperative breeding cichlid from Lake Tanganyika in two learning tests after they experienced an early life programming of the stress axis. During day 10-60 of the fish's development, the offspring were exposed to either i) a cortisol, ii) a mifepristone (a blocker of the glucocorticoid receptors) or iii) a control treatment. Nearly one year later the fish were tested for their learning performance and behavioural flexibility with two learning tests: first a colour discrimination learning test, second a reversal test. In the colour discrimination test fish treated with mifepristone reached the learning criterion quicker than the fish from the control treatment. Further, fish treated with cortisol in their early life required more trials to successfully complete the reversal test compared to the control or mifepristone treated fish.

The authors conclude that early life programming of the stress axis by cortisol or mifepristone influences the learning capability of a highly social cichlid fish. These results together with a previous study, wherein it was shown that the same cortisol treatment led to a reduced social competence in the same species of fish, are evidence that the mechanisms underlying social and non-social flexibility are shaped by the same cognitive mechanisms.

While the search of neuronal mechanisms underlying behavioural flexibility in mammals is well progressed, we still lack a more general understanding of social and non-social flexibility in many other organisms. Therefore, the fascinating results of this well-designed study with a highly social fish as a model species will be of general interest to a wider public.

I have a few, minor comments:

Line 134: The fish used in the experiment were of both sexes. As the treatments were balanced according to sex there is no obligation to investigate a potential sex difference. Nevertheless, I was wondering if the authors ever considered to have a look into potential different abilities between males and females.

Line 142: It is a bit confusing why the authors refer to three tasks here for the first time, as before there were always only two learning tests mentioned. On the first reading it is not clear what is meant by the "motor task". Maybe this sentence can be reformulated to make it clear already here that the "motor task" signifies the training needed to use the experimental apparatus with which the two learning tests are carried out.

Line 157-162: I don't see a problem with the individually flexible time limit based on earlier trials used here. I was just wondering if this procedure reinforces a quick choice of the focal subject on the expense of a higher error rate, which may prolong the training phase.

Line 176: It would be interesting to get some information whether this standardization of the swimming distance by placing the reward always in the row closest to the shelter reduced the error rate and speeded up the choice of the focal fish. Since you used all available rows in the training phase, did you experienced it that the fish was less motivated to search in rows farther from his shelter?

Line 186: You write "The first disc that was touched by a fish (...) was considered as chosen". Two lines later you write "...and this was done as soon as an individual had dislodged the

rewarded disc and had eaten the reward.” Does this mean, that you allowed 5minutes to give the fish the chance to discover the food reward also after a wrong choice?

Line 212: “To reach the learning criterion, a fish had to make at least 10 correct choices in 12 consecutive trials, regardless of non-choice being done.” Does this mean the cases where a fish didn’t choose, the trial was annulated and not counted for the day?
10 out of 12 means that those 12 choices happened always within two blocks and thus two days, but not more?

Line 237: “Colour” did not significantly influence either model, which is unusual, as several fish species showed clear colour preferences in various colour discrimination tests. However, as far as I know, nobody used blue and yellow for *N. pulcher* before.

Table 1: standard length turned out to be a significant factor in the model of the reversal colour discrimination, however you never discuss this result in the text. Couldn’t that be relevant, especially as in fish size is often closely linked with age?

Author's Response to Decision Letter for (RSPB-2021-1289.R0)

See Appendix A.

RSPB-2022-0117.R0

Review form: Reviewer 3

Recommendation

Accept with minor revision (please list in comments)

Scientific importance: Is the manuscript an original and important contribution to its field?

Excellent

General interest: Is the paper of sufficient general interest?

Excellent

Quality of the paper: Is the overall quality of the paper suitable?

Good

Is the length of the paper justified?

Yes

Should the paper be seen by a specialist statistical reviewer?

No

Do you have any concerns about statistical analyses in this paper? If so, please specify them explicitly in your report.

Yes

It is a condition of publication that authors make their supporting data, code and materials available - either as supplementary material or hosted in an external repository. Please rate, if applicable, the supporting data on the following criteria.

Is it accessible?

Yes

Is it clear?

Yes

Is it adequate?

Yes

Do you have any ethical concerns with this paper?

Yes

Comments to the Author

The study sought to test the hypothesis that developmental programming of the HPI stress axis in a species of cichlid fish affects non-social behavioural flexibility in adulthood. The experiment manipulated the early life exposure of cichlid fish to CORT or a CORT-agonist in comparison to a vehicle control, measuring in adulthood performance on a discrimination learning and reversal learning task. CORT treatment reduced reversal learning performance. Combining with prior evidence that CORT treatment reduced social behavioural flexibility (in a separate cohort of the same species), this is to my knowledge a novel demonstration that HPI/HPA axis programming alters both social and non-social behavioural flexibility in the same species. The suggested implication that there is a common cognitive mechanism (involving the HPI axis) underpinning both social and non-social flexibility seems justified by the data, and of significance to a broad readership.

Overall, the experimental design and analysis appear sound and the MS clearly written. The authors have responded constructively to previous reviewer feedback. The addition of minor methodological details would be useful to allow for a future full replication, fulfill ethical requirements as stated by the journal, and potentially allow evaluation of the likely ecological validity (and welfare aspects) of the level of CORT treatment used. There is some scope to draw on related evidence from a wider range of taxa in the Discussion. Personally, I prefer to see marginally-significant results ($0.05 < p < 0.1$) reported as non-significant due to their low evidential value; however the marginal significance (of the effect of CORT-agonist treatment) was clear from the writing (described as a 'tendency') so I suggest the reporting should follow any journal Editorial policy. Additional reporting of the relationship between acquisition of the discrimination and reversal learning performance within individual fish would be a useful minor addition of interest to some readers.

Please see comments below for further details.

Additional Ethical information

The fate of the animals is not indicated as per the Journal policy.

For scientific and welfare reasons, reporting of the concentration of drug treatments is needed (ideally with some information about natural CORT variation in the species).

Information on observer blinding is needed (as per ARRIVE).

Addressing Reduction (e.g., sample size justification) and Refinement (e.g., justification of single-housing of fish) would be an improvement to the reporting.

Were there any differential mortality rates (or growth rates if known) across treatments?

Husbandry information for the early-life period is absent (including stocking density), as well as information about sourcing of animals.

Some of this information could be provided as ESM?

Minor comments:

23: Stress axis - suggest hypothalamic-pituitary-interrenal (HPI) axis at first use for greater clarity

24: mifepristone - useful to state at first use the type of compound this belongs to as antiglucocorticoid/glucocorticoid receptor agonist.

Keywords: +HPA axis? +developmental plasticity?

46-47: reference needed

50: 'as yet', or clearer 'it remains unclear'

62: The argument for looking at behavioural flexibility specifically, rather than there being common effects of HPA on fixed social and non-social behaviours per se, could be strengthened here.

65: I suggest using HPI/HPA stress axis (rather than abbreviated 'stress axis') at the first mention for added clarity.

72: and cognitive flexibility in a non-social domain. [for added clarity]. Please also state what category of compound mifepristone is when first introduced. Additionally, it would be useful to know whether this evidence stated linking MRs, GRs and GCs with cognition refers to cichlids, other fish species, or other vertebrates?

90: For clarity, the end of the Introduction could benefit from a statement of the prediction relating to common pathway versus domain-specific effects.

102-3: Is there any evidence of the fitness benefits of social competence to add to this line of reasoning?

109: Useful to know about the parentage of these 31 broods - were they all unrelated?

110: more information about the drug treatments is needed to allow for a replication (and ethics): 'repeatedly' requires greater explanation - How often? At which ages? How were drugs administered and in what concentration and duration, and with which solvent? This information is needed to ascertain whether cortisol treatment was within a naturalistic range. What housing and husbandry was used during early-life? Were these drug manipulations identical to the treatments in Reyes-Contreras et al. (2019)?

119: The fish were feed five; the fish were fed five [plus useful to state fish flake brand and amount - ad libitum quantities presumably?]

123: 48 fish representing how many of the broods? How were they selected?

129: replace 'experiment' with 'learning tasks'

139-144. Somewhere here the nature of the apparatus requires explanation or referral to the figure, since currently 'three consecutive holes from any of the rows' is unclear - no holes or rows have been mentioned previously.

142: Could the colour shades of the disks be described numerically? If the green shade used in training was more yellow than blue, could this explain later effects of colour on learning performance perhaps?

156: Is there information available on how many training trials were required to reach this stage (or whether there was any treatment difference)?

178: allowed for up to 5 min - delete 'for'

182: And when was the trial terminated if the fish chose the unrewarded disc first? Was it allowed to continue for up to 5 mins still?

190-195: Some more information about how behaviour was observed/scored would be useful for a full replication. Were choices/latencies live-scored by the Experimenter or analysed from videos? Was any intra/inter-observer reliability measure made? Was the Experimenter blind to treatment? Was the Experimenter visible to the fish?

197: reached the learning criterion +(see below)

209: from which treatments/sexes were the 3 fish which failed to meet the learning criterion?

213: +citation for R

253: I'm not convinced that 'feedback' is the clearest term to use here - HPA programming affects ability to cope with stressors, but there is not (direct) evidence shown here of feedback in the reverse direction - coping with stressors influencing HPA programming. The novel aspect here seems to be that early-life stress programming affects the ability to cope with both abiotic and social stressors. It seems previously demonstrated that stress programming affects behaviour(al flexibility) in either context separately.

295-6: There is evidence of a relationship between developmental GC exposure and learning performance (including social learning and reversal learning) in birds, which could be incorporated also. e.g. Crino et al 2014 Anim Behav; Boogert et al 2018 Phil Trans R Soc B; Bebus et al 2016 Anim Behav.

206: reach to the age of

323: Does there not also exist evidence of threat-sensitive foraging/anti-predator behaviour in fish? Given the argument being made is based on the present evidence concerning a fish species, this seems worth mentioning here too with respect to non-social flexibility. It seems incomplete to only refer to penguins to evidence this point, if there is also evidence within the taxa under current study.

331: A missing piece in the argument here seems to be that social competence helps maintain/achieve group membership – can evidence of this be stated?

336: reversal learning speed even negatively – delete ‘even’

347: What may be the advantage... This reads like a new paragraph.

352: different environmental and social cues needs [to add to clarity]

366: cope with both abiotic and social environmental stressors (because the novel part of this study is the simultaneous effect on both; that early-life HPA programming influences the ability to cope with stressors is not so very novel).

Fig 3 legend: Numbered references would be useful for ease of reference here.

Decision letter (RSPB-2022-0117.R0)

01-Apr-2022

Dear Miss Reyes Contreras

I am pleased to inform you that your manuscript RSPB-2022-0117 entitled "Stress axis programming generates long-term effects on cognitive abilities in a cooperative breeder" has been accepted for publication in Proceedings B.

The referee and Associate Editor have recommended publication, but also suggest some minor revisions to your manuscript. Therefore, I invite you to respond to the referee's comments and revise your manuscript. In particular, please see the referee's (and Associate Editor's) concern about the reporting of the non-significant result of treatment: describing this as 'tended', especially in the abstract, can give the impression of an effect, so you need to make it clear that there is no statistical support for treatment differences. Because the schedule for publication is very tight, it is a condition of publication that you submit the revised version of your manuscript within 7 days. If you do not think you will be able to meet this date please let us know.

When submitting your revision please upload a file under "Response to Referees" - in the "File Upload" section. This should document, point by point, how you have responded to the reviewers' and Editors' comments, and the adjustments you have made to the manuscript. We also require a copy of the revised manuscript showing track changes to be uploaded.

4) Data accessibility section and data citation

It is a condition of publication that data supporting your paper are made available either in the electronic supplementary material. Authors must complete the 'data accessibility' section in the submission system. This should list the database and accession number for all data from the article that has been made publicly available, for instance:

NB. From April 1 2013, peer reviewed articles based on research funded wholly or partly by RCUK must include, if applicable, a statement on how the underlying research materials – such as data, samples or models – can be accessed.

[http://datadryad.org/submit?journalID=RSPB&manu=\(Document not available\)](http://datadryad.org/submit?journalID=RSPB&manu=(Document%20not%20available)) which will take you to your unique entry in the Dryad repository. If you have already submitted your data to dryad you can make any necessary revisions to your dataset by following the above link.

Please include the Dryad DOI in the Data Accessibility section and reference in the paper's bibliography.

Please see our Data Sharing Policies (<https://royalsociety.org/journals/authors/author-guidelines/>).

6) A media summary: a short non-technical summary (up to 100 words) of the key findings/importance of your manuscript.

Sincerely,
 Professor Loeske Kruuk
 mailto: proceedingsb@royalsociety.org

Associate Editor
 Comments to Author:

The authors have thoughtfully responded to the referees, and modified the manuscript accordingly. Part of the revision involved reanalysis of the data, substituting length for age, and including colour of the reward in the analyses. The results are very interesting, although one of the key effects of interest in the study dropped out of statistical significance (effect of mifepristone on acquisition of colour discrimination), and is interpreted as a "trend". The referee of the current submission raised this as somewhat problematic, and I agree with their view. In these instances, if one were to ascribe some form of biological significance to a result with a p value > 0.05 (which some would strongly object to, and with fair reason), I think presentation of standardized effect size and confidence interval would be pertinent (i.e. results in interpretation of the effect size not the p value). Notwithstanding, I do not believe the manuscript stands or falls on that one result; there are other interesting causative patterns identified within the study.

The referee has provided a series of very insightful comments that need to be considered and carefully addressed. They have also made an important observation that the fate of the animals is not indicated as required by the journal's policy, and other issues on data reporting in relation to animal and research ethics are also underreported or absent and require clarification. Please attend to these.

I thank the authors for their thorough revision.

Reviewer(s)' Comments to Author:
 Referee: 3
 Comments to the Author(s).

The study sought to test the hypothesis that developmental programming of the HPI stress axis in a species of cichlid fish affects non-social behavioural flexibility in adulthood. The experiment manipulated the early life exposure of cichlid fish to CORT or a CORT-agonist in comparison to a vehicle control, measuring in adulthood performance on a discrimination learning and reversal learning task. CORT treatment reduced reversal learning performance. Combining with prior evidence that CORT treatment reduced social behavioural flexibility (in a separate cohort of the same species), this is to my knowledge a novel demonstration that HPI/HPA axis programming alters both social and non-social behavioural flexibility in the same species. The suggested implication that there is a common cognitive mechanism (involving the HPI axis) underpinning both social and non-social flexibility seems justified by the data, and of significance to a broad readership.

Overall, the experimental design and analysis appear sound and the MS clearly written. The authors have responded constructively to previous reviewer feedback. The addition of minor methodological details would be useful to allow for a future full replication, fulfill ethical requirements as stated by the journal, and potentially allow evaluation of the likely ecological validity (and welfare aspects) of the level of CORT treatment used. There is some scope to draw on related evidence from a wider range of taxa in the Discussion. Personally, I prefer to see marginally-significant results (0.05Please see comments below for further details.

Additional Ethical information

The fate of the animals is not indicated as per the Journal policy.
 For scientific and welfare reasons, reporting of the concentration of drug treatments is needed (ideally with some information about natural CORT variation in the species).

Information on observer blinding is needed (as per ARRIVE).

Addressing Reduction (e.g., sample size justification) and Refinement (e.g., justification of single-housing of fish) would be an improvement to the reporting.

Were there any differential mortality rates (or growth rates if known) across treatments?

Husbandry information for the early-life period is absent (including stocking density), as well as information about sourcing of animals.

Some of this information could be provided as ESM?

Minor comments:

23: Stress axis - suggest hypothalamic-pituitary-interrenal (HPI) axis at first use for greater clarity

24: mifepristone - useful to state at first use the type of compound this belongs to as antiglucocorticoid/glucocorticoid receptor agonist.

Keywords: +HPA axis? +developmental plasticity?

46-47: reference needed

50: 'as yet', or clearer 'it remains unclear'

62: The argument for looking at behavioural flexibility specifically, rather than there being common effects of HPA on fixed social and non-social behaviours per se, could be strengthened here.

65: I suggest using HPI/HPA stress axis (rather than abbreviated 'stress axis') at the first mention for added clarity.

72: and cognitive flexibility in a non-social domain. [for added clarity]. Please also state what category of compound mifepristone is when first introduced. Additionally, it would be useful to know whether this evidence stated linking MRs, GRs and GCs with cognition refers to cichlids, other fish species, or other vertebrates?

90: For clarity, the end of the Introduction could benefit from a statement of the prediction relating to common pathway versus domain-specific effects.

102-3: Is there any evidence of the fitness benefits of social competence to add to this line of reasoning?

109: Useful to know about the parentage of these 31 broods - were they all unrelated?

110: more information about the drug treatments is needed to allow for a replication (and ethics): 'repeatedly' requires greater explanation - How often? At which ages? How were drugs administered and in what concentration and duration, and with which solvent? This information is needed to ascertain whether cortisol treatment was within a naturalistic range. What housing and husbandry was used during early-life? Were these drug manipulations identical to the treatments in Reyes-Contreras et al. (2019)?

119: The fish were feed five; the fish were fed five [plus useful to state fish flake brand and amount - ad libitum quantities presumably?]

123: 48 fish representing how many of the broods? How were they selected?

129: replace 'experiment' with 'learning tasks'

139-144. Somewhere here the nature of the apparatus requires explanation or referral to the figure, since currently 'three consecutive holes from any of the rows' is unclear - no holes or rows have been mentioned previously.

142: Could the colour shades of the disks be described numerically? If the green shade used in training was more yellow than blue, could this explain later effects of colour on learning performance perhaps?

156: Is there information available on how many training trials were required to reach this stage (or whether there was any treatment difference)?

178: allowed for up to 5 min - delete 'for'

182: And when was the trial terminated if the fish chose the unrewarded disc first? Was it allowed to continue for up to 5 mins still?

190-195: Some more information about how behaviour was observed/scored would be useful for a full replication. Were choices/latencies live-scored by the Experimenter or analysed from videos? Was any intra/inter-observer reliability measure made? Was the Experimenter blind to treatment? Was the Experimenter visible to the fish?

197: reached the learning criterion +(see below)

209: from which treatments/sexes were the 3 fish which failed to meet the learning criterion?

213: +citation for R

253: I'm not convinced that 'feedback' is the clearest term to use here - HPA programming affects ability to cope with stressors, but there is not (direct) evidence shown here of feedback in the reverse direction - coping with stressors influencing HPA programming. The novel aspect here seems to be that early-life stress programming affects the ability to cope with both abiotic and social stressors. It seems previously demonstrated that stress programming affects behaviour(al flexibility) in either context separately.

295-6: There is evidence of a relationship between developmental GC exposure and learning performance (including social learning and reversal learning) in birds, which could be incorporated also. e.g. Crino et al 2014 Anim Behav; Boogert et al 2018 Phil Trans R Soc B; Bebus et al 2016 Anim Behav.

206: reach to the age of

323: Does there not also exist evidence of threat-sensitive foraging/anti-predator behaviour in fish? Given the argument being made is based on the present evidence concerning a fish species, this seems worth mentioning here too with respect to non-social flexibility. It seems incomplete to only refer to penguins to evidence this point, if there is also evidence within the taxa under current study.

331: A missing piece in the argument here seems to be that social competence helps maintain/achieve group membership - can evidence of this be stated?

336: reversal learning speed even negatively - delete 'even'

347: What may be the advantage... This reads like a new paragraph.

352: different environmental and social cues needs [to add to clarity]

366: cope with both abiotic and social environmental stressors (because the novel part of this study is the simultaneous effect on both; that early-life HPA programming influences the ability to cope with stressors is not so very novel).

Fig 3 legend: Numbered references would be useful for ease of reference here.

Author's Response to Decision Letter for (RSPB-2022-0117.R0)

See Appendix B.

Decision letter (RSPB-2022-0117.R1)

20-Apr-2022

Dear Miss Reyes Contreras

I am pleased to inform you that your manuscript entitled "Stress axis programming generates long-term effects on cognitive abilities in a cooperative breeder" has been accepted for publication in Proceedings B.

Data Accessibility section

Open Access

Paper charges

Sincerely,

Appendix A

Dear Editor,

We would like to thank you and the two referees for the encouraging feedback. We have done a thorough revision of the manuscript. We expanded the introduction to clarify the broader importance of this research for a general readership, and we added clarifications of methods and results as requested by the referees. Below we provide a point-by-point explanation how we dealt with each comment.

We hope that the revised version of the manuscript will be considered favourably for publication.

*Yours sincerely,
Maria Reyes-Contreras (on behalf of all authors)*

Reviewer(s)' Comments to Author:

Editor

I am writing to inform you that your manuscript RSPB-2021-1289 entitled "Stress axis programming generates long-term effects on cognitive abilities in a cooperative breeder" has, in its current form, been rejected for publication in Proceedings B.

This action has been taken on the advice of referees, who have recommended that substantial revisions are necessary. With this in mind we would be happy to consider a resubmission, provided the comments of the referees are fully addressed. However please note that this is not a provisional acceptance. In particular, you need to make clear the relevance of the study for a broader biological journal such as Proc B, rather than a more specialised journal, and the advances of this particular piece of work (rather than the associated earlier work on the same system); acceptance will be dependent on this.

Thank you very much for this important comment. We revised the manuscript with respect to general implications and – while the introduction is introducing the topic from a broad perspective, we saw that the discussion was indeed lacking the same level of generality. We added now two paragraphs of “General implications” (lines 341-366), which replaces the previous conclusions section, and we added more general interpretations of our results throughout the discussion section (lines 256-260;266-267;288-289;294-296; 297-316).

Referee: 1

Comments to the Author(s)

This manuscript describes an experiment where cichlid fish were exposed in the first two months of life to six doses of either cortisol (the glucocorticoid/stress hormone in fish),

mifepristone (a glucocorticoid blocker), or a control treatment that contained only the carrier of mifepristone. When 18-22 months old, the fish were tested on a colour discrimination and reversal learning task. The authors predicted that the cortisol treatment would impair both colour discrimination and reversal learning performance, while the mifepristone treatment was predicted to enhance performance on both tasks.

The authors found that mifepristone-treated fish performed better on the colour discrimination task, while cortisol-treated fish performed worse on the reversal learning task. Based on these findings and their previous research, the authors conclude that behavioural flexibility in the social (previous study) and non-social domains may share a common underlying cognitive mechanism.

The manuscript is well written and presented and easy to understand. The learning tests follow established protocols, and the results make a significant contribution to the field of animal cognition by suggesting that programming of the HPA axis affects cognitive performance in both social and non-social domains. I enjoyed reading this manuscript and I think the findings are interesting. However, I have some queries regarding the methods, statistical analyses and discussion, which I hope will be helpful to the authors.

We would like to thank the reviewer for the encouraging feedback, and we hope that our explanations below will help to clarify the open methodological points.

Methods:

- If the intensity of facial colour marks is used by these fish to assess conspecific aggressiveness, it's not clear how this relates to distinguishing different colours (rather than colour intensity). Also, if the ecological relevance of the colour discrimination task concerns conspecific interactions, does this task not target performance more closely related to the social domain rather than the non-social domain?

*The referee is correct in mentioning that *N. pulcher* use the intensity of some facial colour marks in the context of aggressiveness. However, the study we had mentioned that was showing this, only manipulated black colour marks (facial stripes). *N. pulcher* can vary the intensity of black facial stripes actively by exposing or retracting the melanophores, which contains the melanin pigment [1,2], but this variability has not been shown or observed in other facial colour marks. As the reference in the paper was only for black facial marks, our statement was possibly confusing and therefore we deleted it.*

*The blue and yellow colour marks are located in different types of chromatophores and they are constituted by different pigmentation types. In particular, the iridescent blue colour in *N. pulcher* facial pattern is a structural colour, which is produced by light reflection. The second colour used in this study (yellow) can be produced either by carotenoid pigments taken up by food or by pteridine pigments [2], and these pigments are not variable over short time spans. Therefore, at least in *N. pulcher*, the intensity of blue and yellow are not adjusted in a social context. Additionally, a previous study reported that *N. pulcher* prefer to interact with yellow-coloured objects in a non-social context, whereas in a social context, when yellow facial marks were artificially enhanced, there was not such preference (ref. [3] now cited also in main text).*

We should also like to mention that, while we did not measure the intensity of the coloured plastic disks used in the learning task, to the human eyes they have fully saturated colours of

highest intensity. In addition, yellow and blue are two colours that have the highest contrast and are known to be recognizable by cichlid fish [3,4]. In the meantime, we have done several learning tasks with all possible colours and combinations, including colours that are not present in body marks (e.g. green, red, purple and white), and we see no differences in the ease they learn the pairs (La Loggia et al. MS). Using yellow and blue were chosen to ensure the fish can see these colours, but in fact it does not seem to matter for the fish which precise colours they have to learn in a colour discrimination task. Because (1) the task is entirely non-social, without social interactions even possible as the focal fish were alone in a tank, (2) the movement of pushing something sideways is not used during any social interaction, and (3) the reward is a food item, we can safely assume that the colour discrimination task is perceived by the fish as non-social task.

- Please describe at what hours of day the learning tasks were conducted, what the inter-trial interval duration was, whether there were any (overnight) breaks between the colour discrimination and reversal learning tasks, whether the task rewards were the only food the fish were provided with, or whether they were starved beforehand or fed extra afterwards.

The training trials as well as the trials of the colour acquisition task and reversal learning were spread over mornings and afternoons. Fish were tested in batches of up to 8 fish. The inter-trial interval was the time the experimenter needed it to set up the experimental apparatus for all test fish. After all fish of a batch received one trial, the next trial was done for all fish. Therefore, the time interval was comparable for all fish on a given experimental day.

Between each block of trials, be it between blocks of the acquisition of colour discrimination or the reversal learning task, there was an overnight break.

The fish were not starved, since the food reward was provided even if they had not done a choice (lines 186-187). In addition, all fish received a few food flakes at the end of each experimental day.

- What was the total duration of social isolation for these fish? Could the fish see each other? If not, does this not represent a stressor?

The total time required for the training, colour acquisition task and reversal task ranged between 9 and 17 days. This time varied because the focal individuals needed different times to reach the learning criterion in the acquisition of colour discrimination and reversal learning tasks. As soon as an individual had passed all tasks, it was immediately returned to its original home tank. Individuals had a unique colour tag, which prevented us for testing the same individual twice.

*Fish were not allowed to see other fish in the adjacent tanks (i) to prevent that fish use social cues from conspecifics to solve a learning task, and (ii) to prevent territorial aggression between individuals, which might have interfered with the training and learning process. *N. pulcher* quickly adopt new tanks with a shelter like it was provided in our experiment, and they defend it vigorously as their territory against conspecifics, even if separated by a glass wall. Such territorial aggression would have greatly disturbed the learning tasks.*

*It is possible that isolation may generate stress, but this is particular true in *N. pulcher* of the smallest size class (<2cm, S. Fischer pers. obs). Here we used larger fish (> 2.7 cm), which*

typically habituate quickly to social isolation, and resume normal activity after 10 or 20 min after entering a new tank. This was also the case in our current experiment, in which all fish got used very readily to their experimental tank and the PVC well plate where the food rewards were provided. Within the habituation phase, all fish started to swim freely in the tank without seeking shelter, even when the experimenter was in the room. They learnt quickly to approach the hand of the experimenter to receive food (see line 187). They actively participated in the tasks, once they had learnt that they are rewarded with krill, which is their favourite food. In fact, the fish became keen in participating, so that after some days they approach the front screen of the tank when the experimenter enters the room waiting for the trial to start. This behaviour contrasts the behaviour of stressed fish, which would remain close to the bottom and hardly move (S. Fischer pers. obs). Moreover, in our fish – besides hiding - stress would have been indicated by dark skin patches, and we never saw such signs in any fish during the entire experiment. So, taken together we do not have any indications that the fish were stressed during our experiment.

- Why was latency measured for only half the test subjects?

We analysed a representative subset of the data (i.e. 24 fish, balanced between colour and sex) and there was no indication that latencies differed between treatments. The significance levels of the model were $P=0.9$ for the cortisol treatment and $P=0.54$ for the mifepristone treatment, when compared to the control treatment, which is far from significant. So even if we had a higher power by using all 48 fish, these results would not change qualitatively.

- How was standard length measured? Was the fish taken out of its tank and placed inside a plastic 'envelope', or was standard length measured from photos/videos? Why was it included in some statistical models – what is this thought to control for?

To measure size, a fish was taken out of a single-sex aggregation group and using a framed measuring board. We measured its length between the tip of the snout to the end of the caudal peduncle. This procedure is very fast, and we ensured that the fish's head and gills remain in a water bubble during the entire measurement to avoid stress. This procedure was done only once before the start of the training phase.

*Although, standard length is important for social interactions in our species, in particular in rank-related interactions, we decided to substitute standard length by age in the cox models. We followed advice of reviewer 2, who had correctly pointed out that age (which is broadly related to size in our species) may be more relevant than size to consider in learning experiments as it may better reflect cognitive development. We totally agree with this argument, in particular as this test is in a non-social context, where size itself should not be so important than during social interactions. Moreover, also in a previous learning study age affected learning abilities of *N. pulcher* [5]. As we know the age of each individual exactly (line 136-138), we replaced size by age in our analyses throughout. The corresponding corrections can be seen in lines 223 and 227.*

Statistical Analyses:

- It is unclear why the statistical models of the colour discrimination, reversal learning and latency data all contain different effects; for example, the main text suggests that only the third, latency, model contains 'family of origin' as a random effect, although the tables in the

main text and the supplementary materials suggest that ‘family’ was included in only the first two models.

-Given that many of the test subjects are from the same broods, it seems that ‘family of origin’ should be included as a random effect in each statistical model? Similarly, why is ‘standard length’ included in only the reversal learning model? Also, it seems that sex should be included as a fixed effect in all models?

In the cox regression proportion hazard models the term frailty [6] (i.e. “family of origin”) was included to account for potential social-group-of-origin effects. This term is indeed included in both models, the colour acquisition task and reversal learning task. The referee is correct in highlighting that inclusion of this factor is only visible in table 1. This is because the output of our statistical software (R) does not provide the results for frailty when testing for the statistical assumptions of the cox-hazard model (Table 2). We included an explanation of ‘frailty’ in line 216-217, where we explain that this corresponds to a random effect as it is usually given in GLMs.

Thank you for pointing out the potential learning differences between males and females. We had balanced the sex of the fish across treatments to account for possible sex effects. Moreover, we included sex of the fish in initial models of colour discrimination acquisition and reversal (see lines 223 and 227). We backwards deleted the factor ‘sex’ of the focal fish because it did not significantly influence the number of blocks the fish needed to reach the learning criterion.

- Given that each test subject has a score for both colour discrimination and reversal learning performance, it would be interesting to plot individuals’ raw scores for each task in the same plot, coloured by early-life treatment. One could visualise and test statistically whether individuals’ scores are correlated across tasks (which would support a ‘general cognitive ability’), or whether this is only true for the control group, given the contrasting effects of the different treatments.

*We would like to thank the reviewer for this thoughtful suggestion. We compiled a new figure, which was included in the SI (Figure S1) and at the end of this document for your information). We want to clarify some aspects about the implications of this new figure. Firstly, we have used block as the unit to evaluate learning criterion. We did not evaluate the scores of the single trials, but we rather grouped trials in block of six successive trials in which a choice was made (correct or wrong choice), to assess if an individual reached our learning criterion in a given block (see learning criterion (line 201) in the main text). Hence, rather than showing individual scores per trials in Figure S1, we show the outcome of individual blocks in percent (%). Second, learning to discriminate a pair of colours is a different cognitive process than reversal learning, which test for behavioural flexibility. The learning task tests how quickly an individual can learn and memorize a stimulus-reward contingency in the first place. Behavioural flexibility is an adaptive behaviour to **changes** in stimulus-reward contingency [7]. Hence, the two processes are expected to be based on different cognitive mechanism; therefore, we do not expect them to be positively correlated.*

Discussion:

- Please discuss your finding that longer fish appear to require significantly more trials to reach the learning criterion.

Following the comment of reviewer 2, we replaced size by age in our models now. Age negatively influences the likelihood to reach the learning criterion in the reversal task. See further details in the section statistical analyses line 223 and 227. We present this new result in the results section, line 239-240, and in Table 1, and discuss it in line 299-307.

- You predicted that performance on both tasks would be affected by both your treatments. Instead, you found that each treatment affected performance on only one of the tasks, but not the other. Why might this be the case? Please discuss.

After following the suggestion of reviewer 2 to replace size by age of the fish, the previously significant effect of mifepristone on performance in the colour discrimination acquisition task turned into a non-significant trend. We included the changes in the result section (line 235) and discussion section (line 274-276), where we now discuss this trend cautiously.

Referee: 2

Comments to the Author(s)

Review:

Stress axis programming generates long-term effects on cognitive abilities in a cooperative breeder

In this study the authors investigate the performance of a cooperative breeding cichlid from Lake Tanganyika in two learning tests after they experienced an early life programming of the stress axis. During day 10-60 of the fish's development, the offspring were exposed to either i) a cortisol, ii) a mifepristone (a blocker of the glucocorticoid receptors) or iii) a control treatment. Nearly one year later the fish were tested for their learning performance and behavioural flexibility with two learning tests: first a colour discrimination learning test, second a reversal test. In the colour discrimination test fish treated with mifepristone reached the learning criterion quicker than the fish from the control treatment. Further, fish treated with cortisol in their early life required more trials to successfully complete the reversal test compared to the control or mifepristone treated fish.

The authors conclude that early life programming of the stress axis by cortisol or mifepristone influences the learning capability of a highly social cichlid fish. These results together with a previous study, wherein it was shown that the same cortisol treatment led to a reduced social competence in the same species of fish, are evidence that the mechanisms underlying social and non-social flexibility are shaped by the same cognitive mechanisms. While the search of neuronal mechanisms underlying behavioural flexibility in mammals is well progressed, we still lack a more general understanding of social and non-social flexibility in many other organisms. Therefore, the fascinating results of this well-designed study with a highly social fish as a model species will be of general interest to a wider public.

We would also like to thank reviewer 2 for the encouraging feedback. We hope that our explanations below will help to clarify the open points

I have a few, minor comments:

Line 134: The fish used in the experiment were of both sexes. As the treatments were balanced according to sex there is no obligation to investigate a potential sex difference.

Nevertheless, I was wondering if the authors ever considered to have a look into potential different abilities between males and females.

In this study, we focused on the long-term effects of early life programming on cognitive performance. All siblings were exposed to the treatments at the same time at an age where sexes are not yet differentiated. Therefore, we do not assume that at this early life stage sex can affect early life programming of the stress axis. Nevertheless, we balanced the sex of the individuals across treatments to account for potential sex effects by our design, and in addition, we also tested for sex effects. Sex was included in the initial models of the acquisition and of the reversal learning task, but we backwards removed sex from both models as it had no significant effect (lines 224-225 and 228-229). Also in all our previous learning experiments with this species no sex effects were detected ([5] and, La Loggia et al. MS).

Line 142: It is a bit confusing why the authors refer to three tasks here for the first time, as before there were always only two learning tests mentioned. On the first reading it is not clear what is meant by the “motor task”. Maybe this sentence can be reformulated to make it clear already here that the “motor task” signifies the training needed to use the experimental apparatus with which the two learning tests are carried out.

Thanks for pointing this out. We added further explanations in line 133-137.

Line 157-162: I don't see a problem with the individually flexible time limit based on earlier trials used here. I was just wondering if this procedure reinforces a quick choice of the focal subject on the expense of a higher error rate, which may prolong the training phase.

In line 153, we included a description the use of two green discs for training purposes (i.e. training task). We added a sentence (line 147-148) to clarify that during the training phase fish had access to food rewards under both green discs, because none of the two discs was blocked. Hence, it was not possible for the fish to make errors in the training phase: both discs were the same colour, and both were rewarded.

Line 176: It would be interesting to get some information whether this standardization of the swimming distance by placing the reward always in the row closest to the shelter reduced the error rate and speeded up the choice of the focal fish. Since you used all available rows in the training phase, did you experienced it that the fish was less motivated to search in rows farther from his shelter?

We did not collect information about the motivation of the fish to search in rows farther away from their shelter in the training phase. However, we have observed in the learning trials that fish inspect the entire plate in search of food, before and/or after making their choice of a disk.

Line 186: You write “The first disc that was touched by a fish (...) was considered as chosen”. Two lines later you write “...and this was done as soon as an individual had dislodged the rewarded disc and had eaten the reward.” Does this mean, that you allowed 5minutes to give the fish the chance to discover the food reward also after a wrong choice?

Yes, this means we allowed 5 min for the food item to be discovered and consumed. This method was also used previously by Buechel et al. 2018 [8] (in our study, we followed their protocol).

Line 212: “To reach the learning criterion, a fish had to make at least 10 correct choices in 12 consecutive trials, regardless of non-choice being done.” Does this mean the cases where a fish didn’t choose, the trial was annulated and not counted for the day? 10 out of 12 means that those 12 choices happened always within two blocks and thus two days, but not more?

Thanks for pointing this out. We agree this may have been confusing. Two blocks always consist of 12 trials with choice taking place. Therefore, two blocks may have extended over more than two days. If no-choice trials occurred, they were not counted when evaluating if a fish reached our learning criterion, and we provided more trials to the fish to compensate for the non-choice trials. We included further explanations in the text in line 204-209 to clarify this methodological procedure.

Line 237: “Colour” did not significantly influence either model, which is unusual, as several fish species showed clear colour preferences in various colour discrimination tests. However, as far as I know, nobody used blue and yellow for *N. pulcher* before.

*Following the suggestion of reviewer 2 we made changes in both cox proportional hazard models. We exchanged the factor size by age and our results had changed, see more details the result section in line 233. When including age instead of size, the factor ‘Colour’ was significant in both models (Table 1). Accordingly, we included some lines of explanation for the preference of yellow colour in *N. pulcher* occurring in both tasks in the discussion section line 309-316.*

Table 1: standard length turned out to be a significant factor in the model of the reversal colour discrimination, however you never discuss this result in the text. Couldn’t that be relevant, especially as in fish size is often closely linked with age?

The reviewer is correct in pointing out the possible importance of age in cognitive processes such as learning. We want to thank the reviewer for this important comment. In both models, the acquisition of colour discrimination and the reversal of colour discrimination we have now substituted standard length by age (see line 223 and 227). We included the results in line 239-240 and discuss the implication of the significant negative effect of age in the reversal-learning task (see line 299-307).

New Figure S1 added to the supplement in response to comment by reviewer 1:

Figure S1. Percent of correct choices in each individual block. The percentages of correct choices (ranging from 0 to 100) are depicted by different symbols as explained in the figure legend. The fish were considered to have learned when they had either 80 (filled circle) or 100 (filled diamond) percent of correct choices in each of the two consecutive blocks (see lines 204-205 in the main text). The treatments are colour coded: control in black, cortisol in red, and mifepristone in blue.

References

1. Balzarini V, Taborsky M, Villa F, Frommen JG. 2017 Computer animations of color markings reveal the function of visual threat signals in *Neolamprologus pulcher*. *Current Zoology* **63**, 45–54. (doi:10.1093/cz/zow086)
2. Luo M, Lu G, Yin H, Wang L, Atuganile M, Dong Z. 2021 Fish pigmentation and coloration: Molecular mechanisms and aquaculture perspectives. *Reviews in Aquaculture* **13**, 2395–2412. (doi:10.1111/raq.12583)
3. Culbert BM, Talagala S, Barnett JB, Stanbrook E, Smale P, Balshine S. 2020 Context-dependent consequences of color biases in a social fish. *Behavioral Ecology* **31**, 1410–1419. (doi:10.1093/beheco/araa099)
4. Escobar-Camacho D, Taylor MA, Cheney KL, Green NF, Justin Marshall N, Carleton KL. 2019 Color discrimination thresholds in a cichlid fish: *Metriaclima benetos*. *Journal of Experimental Biology* **222**. (doi:10.1242/jeb.201160)
5. Bannier F, Tebbich S, Taborsky B. 2017 Early experience affects learning performance and neophobia in a cooperatively breeding cichlid. *Ethology* **123**, 712–723. (doi:10.1111/eth.12646)
6. Landes J, Engelhardt SC, Pelletier F. 2020 An introduction to event history analyses for ecologists. *Ecosphere* **11**, e03238. (doi:10.1002/ecs2.3238)
7. Audet J-N, Lefebvre L. 2017 What's flexible in behavioral flexibility? *Behavioral Ecology* **28**, 943–947. (doi:10.1093/beheco/arx007)
8. Buechel SD, Boussard A, Kotrschal A, van der Bijl W, Kolm N. 2018 Brain size affects performance in a reversal-learning test. *Proceedings of the Royal Society of London B: Biological Sciences* **285**.

Appendix B

Dear Prof. Kruuk, dear Associate Editor,

We were very happy to learn that our paper is accepted pending minor revisions, and we would like to thank the associated editor and the referee for the valuable comments on our manuscript. We have done the corresponding changes in the abstract and result sections, and we provide effect sizes and confidence intervals of the factors entering our models in a new table as it was suggested. Furthermore, we revised the manuscript and included further clarification in response to the comments by the referee. Below we provide a point-by-point explanation how we dealt with each comment.

We hope that the revised version of the manuscript has improved in clarity, and you now find it suitable for final acceptance in the Proceedings of the Royal Society B.

*Yours sincerely,
Maria Reyes-Contreras (on behalf of all authors)*

Associate Editor

Comments to Author:

The authors have thoughtfully responded to the referees and modified the manuscript accordingly. Part of the revision involved reanalysis of the data, substituting length for age, and including colour of the reward in the analyses. The results are very interesting, although one of the key effects of interest in the study dropped out of statistical significance (effect of mifepristone on acquisition of colour discrimination), and is interpreted as a "trend". The referee of the current submission raised this as somewhat problematic, and I agree with their view. In these instances, if one were to ascribe some form of biological significance to a result with a p value > 0.05 (which some would strongly object to, and with fair reason), I think presentation of standardized effect size and confidence interval would be pertinent (i.e. results in interpretation of the effect size not the p value). Notwithstanding, I do not believe the manuscript stands or falls on that one result; there are other interesting causative patterns identified within the study.

Authors' response: As requested we present effect sizes and their confidence intervals in a new table (ESM Table S3), allowing for a biological interpretation of the marginal p -value of 0.06 of the mifepristone treatment in the colour acquisition task, which we also provide in the results section.

The referee has provided a series of very insightful comments that need to be considered and carefully addressed. They have also made an important observation that the fate of the animals is not indicated as required by the journal's policy, and other issues on data reporting in relation to animal and research ethics are also underreported or absent and require clarification. Please attend to these.

Authors' response: We are grateful for these comments and added the information requested by the referee. Please see our point-by-point explanations below.

I thank the authors for their thorough revision.

Reviewer(s)' Comments to Author:

Referee: 3

The study sought to test the hypothesis that developmental programming of the HPI stress axis in a species of cichlid fish affects non-social behavioural flexibility in adulthood. The experiment manipulated the early life exposure of cichlid fish to CORT or a CORT-agonist in comparison to a vehicle control, measuring in adulthood performance on a discrimination learning and reversal learning task. CORT treatment reduced reversal learning performance. Combining with prior evidence that CORT treatment reduced social behavioural flexibility (in a separate cohort of the same species), this is to my knowledge a novel demonstration that HPI/HPA axis programming alters both social and non-social behavioural flexibility in the same species. The suggested implication that there is a common cognitive mechanism (involving the HPI axis) underpinning both social and non-social flexibility seems justified by the data, and of significance to a broad readership.

Overall, the experimental design and analysis appear sound and the MS clearly written. The authors have responded constructively to previous reviewer feedback. The addition of minor methodological details would be useful to allow for a future full replication, fulfill ethical requirements as stated by the journal, and potentially allow evaluation of the likely ecological validity (and welfare aspects) of the level of CORT treatment used. There is some scope to draw on related evidence from a wider range of taxa in the Discussion. Personally, I prefer to see marginally-significant results (0.05 Please see comments below for further details.

*Authors' response: We would like to thank the referee for their thoughtful comments on the ethical section of the manuscript and the minor comments. Accounting for these comments greatly improved the clarity and readability of the manuscript. Given the limitation in the number of pages (the manuscript already has 10 pages, which is the maximum allowed by the journal) we have included most requested information (blinding, reduction, refinement, natural ranges of CORT plasma level in *Neolamprologus pulcher*, fate of the animals, information about mortality rates) in the ESM, but important information such as drug concentrations, refinement, and stock densities of aggregations are given in the main text.*

We assume that one reason for asking for details of the rearing procedure of the experimental fish came from the misconception that the rearing treatment was done during this study. This is not what happened. The fish were reared already during the study by Reyes et al 2019[1], which provided full detail of the procedures, and which were already reviewed during the publication process of our 2019 paper [1]. Some fish of those broods were used during the experiments of [1], and different fish of these broods were used in the current study. So all fish used in this study were entirely naïve to any experimental tests, and after the early rearing were kept in conspecific aggregations, as described in the manuscript, until the experiment of our current study started. This is the reason why we feel the details of the rearing phase do not have to be given in detail in this manuscript. We now made it clear in the main text and the ESM that these fish were reared during our previous study, moreover we give more detail now on the rearing methods in the ESM. As the manuscript increased in length during our revision and we are limited by the maximum page range of the journal, we feel it is justified to keep the methods section on the early life treatments short.

Additional Ethical information

The fate of the animals is not indicated as per the Journal policy.

Authors' response: This information was included in the ESM in line 46 to 49.

Here is what we included in ESM: "At the end of the study, fish from the control treatment were integrated in the breeding stock of our aquarium facility at the Ethologische Station Hasli. All individuals treated with cortisol and mifepristone were sacrificed in accordance with the regulations of our animal facility and the Veterinary Office of the Kanton Bern, Switzerland, licence no. 93/18."

For scientific and welfare reasons, reporting of the concentration of drug treatments is needed (ideally with some information about natural CORT variation in the species).

Authors' response: The concentration of the drug treatment is now included in the main manuscript in the Methods section "Early-life treatments" line 114/115.

*Additional information about natural variation in cortisol levels was included in ESM in line 20/21. We wrote: "In *N. pulcher* adult the mean baseline cortisol plasma levels range from 20-35 ng ml⁻¹ [2] and after an acute stressor plasma cortisol level reaches 500 ng ml⁻¹ [2]"*

Information on observer blinding is needed (as per ARRIVE).

Authors' response: Information about blinding of the observer (MRC) before, during and after the experiment is now reported in SEM lines 40/41 and 43 to 45.

Addressing Reduction (e.g., sample size justification) and Refinement (e.g., justification of single-housing of fish) would be an improvement to the reporting.

*Authors' response: A refinement statement is given in main manuscript in line 128 to 131. Temporary isolation of the fish was necessary to prevent interferences of the training and learning procedures. While therefore the experimental fish were not able to see fish in the adjacent tanks being part of the experiment, there were able to see groups of *N. pulcher* kept in the opposite rack for a different study. At no time point the experimental fish showed any sign of stress (see main text).*

A statement for reduction was included in SEM line 38 to 40. Our sample sizes (16 per treatment) are the minimum possible for this type of experiments to allow for a sufficient statistical power to detect effects that are present. In previous similar studies in our study species, the sample sizes per treatment have been 14 [3] and 20 [4].

Were there any differential mortality rates (or growth rates if known) across treatments?

Authors' response: No, there were no differences. This information was included in ESM in line 37. We wrote: "After early-life treatment, mortality rates and growth rates did not differ between treatments."

Husbandry information for the early-life period is absent (including stocking density), as well as information about sourcing of animals.

Authors' response: Thank you for pointing this out, we included more information about animal sourcing in ESM line 12 to 14, experimental licence, and stock densities during early life period in the main text lines 136 to 139, and line 116.

Some of this information could be provided as ESM?

We explained above which information is now included in the main text and which went to the ESM.

Minor comments:

23: Stress axis – suggest hypothalamic–pituitary–interrenal (HPI) axis at first use for greater clarity

Authors' response: We did the corresponding change in line 23-24.

24: mifepristone - useful to state at first use the type of compound this belongs to as antiglucocorticoid/glucocorticoid receptor agonist.

Keywords: +HPA axis? +developmental plasticity?

Authors' response:

We added 'glucocorticoid-receptor antagonist' in line 24. Regarding, the referee's suggestion of adding more key words; unfortunately, it is not possible to add more key words because the manuscript already contains the maximum number permitted by the Journal (i.e., 6 key words)

46-47: reference needed

Authors' response: We added the corresponding reference in line 47.

50: 'as yet', or clearer 'it remains unclear'

Authors' response: We did the corresponding change in line 51.

62: The argument for looking at behavioural flexibility specifically, rather than there being common effects of HPA on fixed social and non-social behaviours per se, could be strengthened here.

Author's response: We adjusted the sentence in line 60/61

65: I suggest using HPI/HPA stress axis (rather than abbreviated 'stress axis') at the first mention for added clarity.

Authors' response: We did the corresponding change in line 62

72: and cognitive flexibility in a non-social domain. [for added clarity]. Please also state what category of compound mifepristone is when first introduced. Additionally, it would be useful to know whether this evidence stated linking MRs, GRs and GCs with cognition refers to cichlids, other fish species, or other vertebrates?

Authors' response: We made the suggested addition and added the requested clarifications about the receptor blocker (line 73 and 74) and the taxonomic groups studied in the references we cited here (lines 71/72).

90: For clarity, the end of the Introduction could benefit from a statement of the prediction relating to common pathway versus domain-specific effects.

Authors' response: We added the prediction in line 92 to 94

102-3: Is there any evidence of the fitness benefits of social competence to add to this line of reasoning?

Authors' response: We clarified that the higher probability to be accepted by dominants in a group by more socially competent individuals IS a major fitness benefit, because survival depends on acceptance, and we added the respective references in line 106/107.

109: Useful to know about the parentage of these 31 broods - were they all unrelated?

Authors' response: We choose breeding pairs from the stock population at the Ethologische Station Hasli to generate the broods of individuals used in this experiment. Each breeding pair produced only one brood in our experimental set up. Hence, to the best of our knowledge, breeder pairs are unrelated, and thus also the broods are unrelated. As we explained above, because the rearing was done during our previous study we present the details of rearing in the ESM, where we also mention that broods were unrelated (ESM line 29).

110: more information about the drug treatments is needed to allow for a replication (and ethics): 'repeatedly' requires greater explanation - How often? At which ages? How were drugs administered and in what concentration and duration, and with which solvent? This information is needed to ascertain whether cortisol treatment was within a naturalistic range. What housing and husbandry was used during early-life? Were these drug manipulations identical to the treatments in Reyes-Contreras et al. (2019)?

Authors' response: As explained above, we used the fish reared by Reyes-Contreras et al., 2019 [1]. In this study we used siblings of the broods reared in the earlier study that had not undergone any behavioural tests in their life until we tested them for learning and reversal learning. These fish were kept in aggregations in 200-L compartments (as described in the main text line 115 to 117) until we started the learning/reversal learning experiment and have not been used for any experimental tests before. So, the drug manipulations are indeed identical, and they were done only once –we just used sibling fish of the ones used during [1]. We say this now explicitly in the main text (line 113) and in the ESM (line 23/24). This is also the reason why we did not provide all details here. The exact procedures have been reviewed and published. Given the page number limitation of this journal, we included some additional information to address the referee requests (see ESM lines 23 to 35), but for further details we still refer to Reyes-Contreras et al., 2019.

119: The fish were feed five; the fish were fed five [plus useful to state fish flake brand and amount – ad libitum quantities presumably?

Authors' response: Thanks for spotting the typo, it was corrected. We added the brand (SEM line 17/18). We did not feed the fish ad libitum, but according to the feeding regulations in our animal facility (i.e., the amount of food that is eaten completely in 5 min).

123: 48 fish representing how many of the broods? How were they selected?

Authors' response: These were 3 x 8 broods, and we selected one male and one female per brood from an intermediate size range of a given brood (not all siblings grow exactly the same rate in fish).

129: replace 'experiment' with 'learning tasks'

Authors' response: Done

139-144. Somewhere here the nature of the apparatus requires explanation or referral to the figure, since currently 'three consecutive holes from any of the rows' is unclear – no holes or rows have been mentioned previously.

Authors' response: We included the numbers of rows and holes per row in line 132, which is the place where we first mentioned the apparatus, that refers to the number of holes in each row.

142: Could the colour shades of the disks be described numerically? If the green shade used in training was more yellow than blue, could this explain later effects of colour on learning performance perhaps?

*Authors' response: Unfortunately, neither the nuance nor shade of the coloured discs were assessed quantitatively. However, tests with different brands of discs were done in labs in Canada [5] and Austria [4] in *N. pulcher*, and a preference for yellow was always found regardless of whether it was tested against a blue or a red disc, suggesting it is a general preference in our study species.*

156: Is there information available on how many training trials were required to reach this stage (or whether there was any treatment difference)?

Authors' response: The fish need a maximum of 14 trials to continue in the acquisition of colour discrimination task. The experimenter was blind to the treatment during the learning tasks; hence differences between treatments in the training phase were not recorded.

178: allowed for up to 5 min – delete 'for'

Authors' response: Deleted

182: And when was the trial terminated if the fish chose the unrewarded disc first? Was it allowed to continue for up to 5 mins still?

Authors' response: Yes, if there was a wrong choice, we waited up to 5 min to allow the fish to dislodge the correct disc and eat the food reward. We explained this in the next sentence, which reads: "In the few cases, in which a fish did not eat the food reward within 5 min (i.e.,

the fish had made a wrong choice or no choice, but did not uncover the reward), the rewarded disc was moved to open the hole halfway.”

190-195: Some more information about how behaviour was observed/scored would be useful for a full replication. Were choices/latencies live-scored by the Experimenter or analysed from videos? Was any intra/inter-observer reliability measure made? Was the Experimenter blind to treatment? Was the Experimenter visible to the fish?

Authors' response: Thank you for pointing this out, indeed we should have included this information before. We describe the procedures now in SEM lines 42 to 44. Furthermore, there was no intra- or inter-observer reliability measures done, because there was only one observer, and the behavioural measures (first touch of a disc) are very clearly identifiable and allow no room for interpretation. Moreover, we can exclude bias because the observer was blind to the treatment when coding the behaviour.

197: reached the learning criterion +(see below)

Authors' response: We do not understand this comment. Maybe something is missing here. The respective sentence reads "After two consecutive blocks had passed, the learning criterion was assessed." and we do not see that anything may be unclear here.

209: from which treatments/sexes were the 3 fish which failed to meet the learning criterion?

Authors' response: There were 3 females from cortisol treatment

213: +citation for R

Authors' response: Done, the reference was included in line 216

253: I'm not convinced that 'feedback' is the clearest term to use here - HPA programming affects ability to cope with stressors, but there is not (direct) evidence shown here of feedback in the reverse direction - coping with stressors influencing HPA programming. The novel aspect here seems to be that early-life stress programming affects the ability to cope with both abiotic and social stressors. It seems previously demonstrated that stress programming affects behaviour(al flexibility) in either context separately.

Authors' response: The referee is correct in mentioned that we only showed one direction of the feedback. We corrected the sentence (line 261).

295-6: There is evidence of a relationship between developmental GC exposure and learning performance (including social learning and reversal learning) in birds, which could be incorporated also. e.g. Crino et al 2014 Anim Behav; Boogert et al 2018 Phil Trans R Soc B; Bebus et al 2016 Anim Behav.

Authors' response: We would like to thank the referee to point out these references to us, which indeed nicely illustrate how corticosteroid levels modulate non-social learning (Crino et al 2014) and social learning (Boogert et al 2018 Phil Trans R Soc B) and how early life pharmacological administration of corticosteroids influences learning later in life (Bebus et al 2016 Anim Behav). Yet, our argument in line 301 to 303 is about the role of glucocorticoid receptors (not GCs) in social and non-social behaviour flexibility. We mentioned the link

between GCs, or more generally, the stress system, and cognition/learning in the introduction (line 66). Because here we make a specific statement about the receptors and we talk about flexibility, we rather feel that the inclusion of such references may be misleading in this place.

206: reach to the age of

Authors' response: Thanks. We inserted "to" (line 312)

323: Does there not also exist evidence of threat-sensitive foraging/anti-predator behaviour in fish? Given the argument being made is based on the present evidence concerning a fish species, this seems worth mentioning here too with respect to non-social flexibility. It seems incomplete to only refer to penguins to evidence this point, if there is also evidence within the taxa under current study.

*Authors' response: In response to this comment, we exchanged the example of penguins by an example of behavioural adjustment under high environmental risk in the fish species *Pseudorasbora parva* (line 326 to 328).*

331: A missing piece in the argument here seems to be that social competence helps maintain/achieve group membership – can evidence of this be stated?

Authors' response: Yes, they are more likely accepted. We included the corresponding reference for the evidence that more socially competent individuals are more likely to be accepted in a group (lines 334/335).

336: reversal learning speed even negatively – delete 'even'

Authors' response: Done, thank you.

347: What may be the advantage... This reads like a new paragraph.

Authors' response: We see the point, but we would prefer to retain this as one paragraph discussing the existence and possible trade-offs linked to a shared cognitive mechanism; and to contrast this with our second general conclusion (second paragraph), which is about stress programming underlying flexibility.

352: different environmental and social cues needs [to add to clarity]

Authors' response: To our understanding social cues are part of the environmental cues, so whenever we wrote "environmental" we in fact referred to all cues from the environment, social and non-social, so we prefer not to change this in this sentence.

366: cope with both abiotic and social environmental stressors (because the novel part of this study is the simultaneous effect on both; that early-life HPA programming influences the ability to cope with stressors is not so very novel).

Authors' response: We changed the sentence for further clarification (lines 367 to 370).

Fig 3 legend: Numbered references would be useful for ease of reference here.

Authors' response: We changed the references to numbered references.

References

1. Reyes-Contreras M, Glauser G, Rennison DJ, Taborsky B. 2019 Early-life manipulation of cortisol and its receptor alters stress axis programming and social competence. *Philosophical Transactions of the Royal Society B: Biological Sciences* **374**. (doi:10.1098/rstb.2018.0119)
2. Mileva VR, Fitzpatrick JL, Marsh-Rollo S, Gilmour KM, Wood CM, Balshine S. 2009 The Stress Response of the Highly Social African Cichlid *Neolamprologus pulcher*. *Physiological and Biochemical Zoology* **82**, 720–729. (doi:10.1086/605937)
3. Bannier F, Tebbich S, Taborsky B. 2017 Early experience affects learning performance and neophobia in a cooperatively breeding cichlid. *Ethology* **123**, 712–723. (doi:10.1111/eth.12646)
4. Fischer S, Balshine S, Hadolt MC, Schaedelin FC. 2021 Siblings matter: Family heterogeneity improves associative learning later in life. *Ethology* **127**, 897–907. (doi:10.1111/eth.13196)
5. Culbert BM, Talagala S, Barnett JB, Stanbrook E, Smale P, Balshine S. 2020 Context-dependent consequences of color biases in a social fish. *Behavioral Ecology* **31**, 1410–1419. (doi:10.1093/beheco/araa099)